# QUERY-POLICY MISALIGNMENT IN PREFERENCE-BASED REINFORCEMENT LEARNING

**Xiao Hu**[1*]**, Jianxiong Li**[1*]**, Xianyuan Zhan**[1,2†]**, Qing-Shan Jia**[1†] **& Ya-Qin Zhang**[1†]
[1] Tsinghua University, Beijing, China    [2] Shanghai Artificial Intelligence Laboratory, Shanghai, China
{hu-x21,li-jx21}@mails.tsinghua.edu.cn, zhanxianyuan@air.tsinghua.edu.cn

## ABSTRACT

Preference-based reinforcement learning (PbRL) provides a natural way to align RL agents' behavior with human desired outcomes, but is often restrained by costly human feedback. To improve feedback efficiency, most existing PbRL methods focus on selecting queries to maximally improve the overall quality of the reward model, but counter-intuitively, we find that this may not necessarily lead to improved performance. To unravel this mystery, we identify a long-neglected issue in the query selection schemes of existing PbRL studies: *Query-Policy Misalignment*. We show that the seemingly informative queries selected to improve the overall quality of reward model actually may not align with RL agents' interests, thus offering little help on policy learning and eventually resulting in poor feedback efficiency. We show that this issue can be effectively addressed via *policy-aligned query* and a specially designed *hybrid experience replay*, which together enforce the bidirectional query-policy alignment. Simple yet elegant, our method can be easily incorporated into existing approaches by changing only a few lines of code. We showcase in comprehensive experiments that our method achieves substantial gains in both human feedback and RL sample efficiency, demonstrating the importance of addressing *query-policy misalignment* in PbRL tasks. Code is available at https://github.com/huxiao09/QPA.

## 1 INTRODUCTION

Reward plays an imperative role in every reinforcement learning (RL) problem. It specifies the learning objective and incentivizes agents to acquire correct behaviors. With well-designed rewards, RL has achieved remarkable success in solving many complex tasks (Mnih et al., 2015; Silver et al., 2017b; Degrave et al., 2022). However, designing a suitable reward function remains a longstanding challenge (Abel et al., 2021; Li et al., 2023; Sorg, 2011). Due to human cognitive bias and system complexity (Hadfield-Menell et al., 2017), it is difficult to accurately convey complex behaviors through numerical rewards, resulting in unsatisfactory or even hazardous agent behaviors.

Preference-based RL (PbRL), also known as RL from human feedback (RLHF), promises learning reward functions autonomously without the need for tedious hand-engineered reward design (Christiano et al., 2017; Lee et al., 2021a;b; Park et al., 2022; Liang et al., 2022; Shin et al., 2023; Tien et al., 2023). Instead of using provided rewards, PbRL queries a (human) overseer to provide preferences between a pair of agent's behaviors, and the RL agent seeks to maximize a reward function that is trained to be consistent with human preferences. This approach provides a more natural way for humans to communicate their desired outcomes to RL agents, enabling more desirable behaviors (Christiano et al., 2017). However, labeling a large number of preference queries requires tremendous human effort, inhibiting its wide application in real-world scenarios (Lee et al., 2021b; Park et al., 2022; Liang et al., 2022). Thus, in PbRL, the feedback efficiency is of utmost importance.

To enable feedback-efficient PbRL, it is crucial to carefully select which behaviors to query the overseer's preference and which ones not to, in order to extract as much information as possible from each preference labeling process (Christiano et al., 2017; Lee et al., 2021b; Biyik & Sadigh, 2018; Biyik et al., 2020). Motivated by this, existing works focus on querying the most "informative"

---

[*]The two authors contributed equally.
[†]Correspondence to Xianyuan Zhan, Qing-Shan Jia and Ya-Qin Zhang

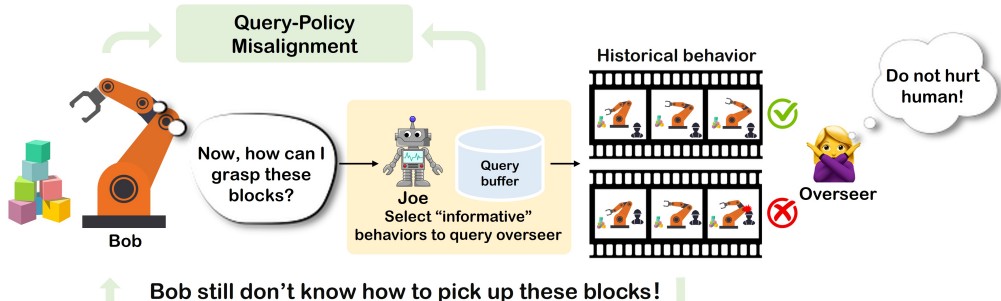

Figure 1: Illustration of *query-policy misalignment*. Bob's current focus is on grasping the blocks. However, the overseer advises him not to cause harm to humans instead of providing guidance on grasping techniques.

behaviors for preferences that are likely to maximally rectify the overall reward model, such as sampling according to ensemble disagreements, mutual information, or behavior entropy (Lee et al., 2021b; Shin et al., 2023). However, it is also observed that these carefully designed query schemes often only marginally outperform the simplest scheme that randomly selects behaviors to query human preferences (Ibarz et al., 2018; Lee et al., 2021b). This counter-intuitive phenomenon brings about a puzzling question: *Why are these seemingly informative queries actually not effective in PbRL training?*

In this paper, we identify a long-neglected issue in the query schemes of existing approaches that is responsible for their ineffectiveness: *Query-Policy Misalignment*. Take Figure 1 as an illustration, Bob is a robot attempting to pick up some blocks, while Joe is a querier selecting the behaviors that he considers the most informative for the reward model to query the overseer. However, the chosen behaviors may not align with Bob's current interests. So, even after the query, Bob is still clueless about how to pick up blocks, which means that a valuable query opportunity might be wasted. In Section 4, we provide concrete experiments and find that such misalignment is prevalent in previous query schemes. Specifically, we observe that the queried behaviors often fall outside the scope of the current policy's visitation distribution, indicating that they are less likely to be encountered by the current RL agent, and thus are not what the current agent's interests. Therefore, the queried behaviors have little impact on the current policy learning and result in poor feedback efficiency.

Interestingly, we find that the *query-policy misalignment* issue can be easily addressed by *policy-aligned query selection*, which can be implemented by making a simple modification to existing query schemes. This technique ensures that the query scheme only selects the recent behaviors of RL agents, which in turn enables the overseer to provide timely feedback on relevant behaviors for policy learning rather than some irrelevant experiences. By making this minimalist modification to the query schemes of existing methods, we showcase substantial improvements in terms of both feedback and sample efficiency as compared to the base schemes. Further leveraging the insight from *query-policy misalignment*, we introduce a simple technique, called *hybrid experience replay*, that simply updates RL agents using experiences uniformly sampled from the entire replay buffer and some recent experiences. Intuitively, this technique ensures that the RL agent updates more frequently on the region where human preferences have been recently labeled, thereby further aligning the policy learning with the recent human preferences.

In summary, the combination of *policy-aligned query selection* and *hybrid experience replay* establishes bidirectional query-policy alignment, making every query accountable for policy learning. Notably, these techniques can be easily incorporated into existing PbRL approaches (Lee et al., 2021b; Park et al., 2022) with minimal modifications. We evaluate our proposed method on benchmark environments in DeepMind Control Suite (DMControl) (Tassa et al., 2018) and MetaWorld (Yu et al., 2020). Simple yet elegant, experimental results demonstrate significant feedback and sample efficiency gains, highlighting the effectiveness of our proposed method and the importance of addressing *query-policy misalignment* in PbRL.

## 2 RELATED WORK

PbRL provides a natural approach for humans (oracle overseer) to communicate desired behaviors with RL agents by making relative judgments between a pair of behaviors (Akrour et al., 2011; Pilarski et al., 2011; Christiano et al., 2017; Stiennon et al., 2020; Wu et al., 2021). However,

acquiring preferences is typically costly, imposing high demands on feedback efficiency (Lee et al., 2021b; Park et al., 2022; Liang et al., 2022).

**Query selection schemes in PbRL.** It is widely acknowledged that the query selection scheme plays a crucial role in PbRL for improving feedback efficiency (Christiano et al., 2017; Biyik & Sadigh, 2018; Biyik et al., 2020; Ibarz et al., 2018; Lee et al., 2021b). Motivated by this idea, prior works commonly assess the information quality of queries using metrics such as entropy (Biyik & Sadigh, 2018; Ibarz et al., 2018; Lee et al., 2021b), the L2 distance in feature space (Biyik et al., 2020) or ensemble disagreement of the reward model (Christiano et al., 2017; Ibarz et al., 2018; Lee et al., 2021b; Park et al., 2022; Liang et al., 2022). Based on these metrics, researchers often employ complex sampling approaches such as greedy sampling (Biyik & Sadigh, 2018), K-medoids algorithm (Biyik & Sadigh, 2018; Rdusseeun & Kaufman, 1987), or Poisson disk sampling (Bridson, 2007; Biyik et al., 2020), *etc.*, to sample the most "informative" queries. Despite adding extra computational costs, it is observed that these complex schemes often benefit little to policy learning, and sometimes perform similarly to the simplest scheme that directly queries humans with randomly selected queries (Lee et al., 2021b; Ibarz et al., 2018). In this paper, we identify a common issue with the existing schemes, *query-policy misalignment*, which shows that the selected seemingly "informative" queries may not align well with the current interests of RL agents, providing an explanation of why existing schemes often lead to less improved feedback efficiency.

**Other techniques for improving feedback efficiency.** In addition to designing effective query selection schemes, there are other efforts to improve the feedback efficiency of PbRL from various perspectives. For instance, some works focus on ensuring that the initial queries are feasible for humans to provide high-quality preferences by initializing RL agents with imitation learning (Ibarz et al., 2018) or unsupervised-pretraining (Lee et al., 2021b). Liang et al. (2022) shows that adequate exploration can improve both sample and feedback efficiency. Recently, Park et al. (2022) applies pseudo-labeling (Lee et al., 2013) in semi-supervised learning along with temporal cropping data augmentation to remedy the limited human preferences and achieves SOTA performances. Note that our proposed method is orthogonal to these techniques and can be used in conjunction with any of them with minimal code modifications.

**Local decision-aware model learning.** The key idea of our proposed method is to learn the reward mode precisely within the distribution of the current policy, rather than inadequately within the entire global state-action space. Some model-based decision-making methods share a similar idea with ours, emphasizing the importance of learning a critical local dynamics model instead of a global one to enhance sample complexity. Specifically, a line of model-based policy search work aims to learn the local dynamics model on current states or trajectories to expedite convergence (Levine & Abbeel, 2014; Levine et al., 2015; Fu et al., 2016; Lioutikov et al., 2014; Bagnell & Schneider, 2001; Atkeson et al., 1997). Some work on value-equivalent models focuses on learning the dynamics model relevant for value estimation (Grimm et al., 2020; Srinivas et al., 2018; Oh et al., 2017; Silver et al., 2017a; Tamar et al., 2016). Several model-based RL methods propose to learn the model centering on the current policy or task (Wang et al., 2023; Lambert et al., 2020; Ma et al., 2023).

## 3 PRELIMINARY

The RL problem is typically specified as a Markov Decision Process (MDP) (Puterman, 2014), which is defined by a tuple $\mathcal{M} := (\mathcal{S}, \mathcal{A}, r, T, \gamma)$. $\mathcal{S}, \mathcal{A}$ represent the state and action space, $r : \mathcal{S} \times \mathcal{A} \to \mathbb{R}$ is the reward function, $T : \mathcal{S} \times \mathcal{A} \to \mathcal{S}$ is the dynamics and $\gamma \in (0, 1)$ is the discount factor. The goal of RL is to learn a policy $\pi : \mathcal{S} \to \mathcal{A}$ that maximizes the expected cumulative discounted reward.

**Off-policy actor-critic RL.** To tackle the high-dimensional state-action space, off-policy actor-critic RL algorithms typically maintain a parametric Q-function $Q_\theta(s, a)$ and a parametric policy $\pi_\phi(a|s)$, which are optimized via alternating between *policy evaluation* (Eq. (1)) and *policy improvement* (Eq. (2)) steps. The *policy evaluation* step seeks to enforce $Q_\theta(s, a)$ to be consistent with the empirical Bellman operator that backs up samples $(s, a, s')$ stored in replay buffer $\mathcal{D}$, while the *policy improvement* step improves $\pi_\phi$ via maximizing the learned Q-value:

$$\hat{Q}^{k+1} \leftarrow \arg\min_Q \mathbb{E}_{s,a,s'\sim\mathcal{D}} \left[ \left( r(s,a) + \gamma \mathbb{E}_{a'\sim\hat{\pi}^k(a'|s')} \left[ \hat{Q}^k \left( s', a' \right) \right] - Q(s,a) \right)^2 \right] \quad (1)$$

$$\hat{\pi}^{k+1} \leftarrow \arg\max_\pi \mathbb{E}_{s\sim\mathcal{D}, a\sim\pi(a|s)} \left[ \hat{Q}^{k+1}(s,a) \right] \quad (2)$$

**Preference-based RL.** Different from the standard RL setting, the reward signal is not available in PbRL. Instead, a (human) overseer provides preferences between pairs of *trajectory segments*,

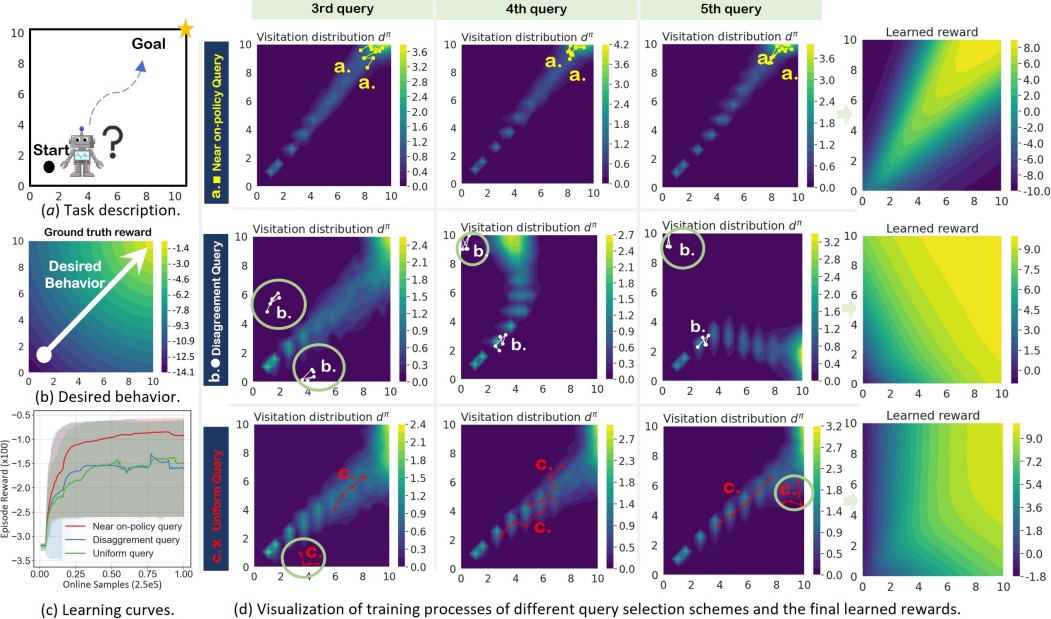

Figure 2: Impacts of *query-policy misalignment* in PbRL training. (a). 2D navigation task. RL agent should navigate to the goal. (b). The desired behavior of this task is to move to the goal in a straight line. (c). The learning curves of different query selection methods. (d). Existing query selection methods often select queries that lie outside the visitation distribution of the current policy.

and the agent leverages these feedbacks to learn a reward function $\widehat{r}_\psi : \mathcal{S} \times \mathcal{A} \to \mathbb{R}$ to be consistent with the provided preferences. A trajectory segment $\sigma$ is a sequence of states and actions $\{s_k, a_k, \cdots, s_{k+L-1}, a_{k+L-1}\} \in (\mathcal{S} \times \mathcal{A})^L$, which is typically shorter than the whole trajectories. Given a pair of segments $(\sigma^0, \sigma^1)$, the overseer provides a feedback signal $y$ indicating which segment the overseer prefer, i.e., $y \in \{0, 1\}$, where 0 indicates the overseer prefers segment $\sigma^0$ over $\sigma^1$, 1 otherwise. Following the Bradley-Terry model (Bradley & Terry, 1952), we can model a preference predictor $P_\psi$ using the reward function $\widehat{r}_\psi(s, a)$:

$$P_\psi \left[ \sigma^1 \succ \sigma^0 \right] = \frac{\exp \sum_t \widehat{r}_\psi \left( s_t^1, a_t^1 \right)}{\sum_{i \in \{0,1\}} \exp \sum_t \widehat{r}_\psi \left( s_t^i, a_t^i \right)} \tag{3}$$

where $\sigma^1 \succ \sigma^0$ denotes the overseer prefers $\sigma^1$ than $\sigma^0$. Typically, $\widehat{r}_\psi$ is optimized by minimizing the cross-entropy loss of preference predictor $P_\psi$ and the true preference label $y$.

$$\mathcal{L}^{\text{reward}} = - \mathop{\mathbb{E}}_{(\sigma^0, \sigma^1, y) \sim \mathcal{D}^\sigma} \left[ (1 - y) \log P_\psi \left[ \sigma^0 \succ \sigma^1 \right] + y \log P_\psi \left[ \sigma^1 \succ \sigma^0 \right] \right] \tag{4}$$

where $\mathcal{D}^\sigma$ denotes the preference buffer which stores the history overseer's preferences $\{(\sigma^0, \sigma^1, y)\}$. Most recent PbRL methods (Lee et al., 2021b; Park et al., 2022; Liang et al., 2022) are built upon off-policy actor-critic RL algorithms to enhance sample and feedback efficiency. In these methods, pairs of segments $(\sigma^0, \sigma^1)$ are selected from trajectories in the off-policy RL replay buffer $\mathcal{D}$. These selected pairs are then sent to the overseer for preference query, yielding feedback $(\sigma^0, \sigma^1, y)$. These feedback instances are subsequently stored in the separate preference buffer $\mathcal{D}^\sigma$ for reward learning. The *query selection scheme* refers to the strategy that decides which pair of segments $(\sigma^0, \sigma^1)$ should be selected from $\mathcal{D}$ for preference query prior to the reward learning. An effective query selection scheme is of paramount importance to achieve high feedback efficiency in PbRL.

## 4 QUERY-POLICY MISALIGNMENT

**A motivating example.** In this section, we conduct an intuitive experiment to demonstrate a prevalent but long-neglected issue: *query-policy misalignment*, which accounts for the poor feedback efficiency of existing query selection schemes. Specifically, as illustrated in Figure 2(a), we consider a 2D continuous space with $(x, y)$ coordinates defined on $[0, 10]^2$. For each step, the RL agent can move $\Delta x$ and $\Delta y$ ranging from $[-1, 1]$. We want the agent to navigate from the start to the goal as quickly as possible. We run PEBBLE (Lee et al., 2021b), a popular PbRL method, with two widely used query selection schemes in previous studies: *uniform query selection* that randomly selects segments to query preferences and *disagreement*

*query selection* that selects the segments with the largest ensemble disagreement of preference predictors (Lee et al., 2021b; Christiano et al., 2017; Ibarz et al., 2018; Park et al., 2022). We also run PEBBLE with *policy-aligned query selection* that selects the recently collected segments from the policy-environment interactions and will be further investigated in later content. We track the distribution $d^\pi$ of current agent's policy $\pi$ throughout the training process and plot the selected segments of different query selection schemes in Figure 2(d). Please refer to the Appendix C.1 for detailed experimental setups.

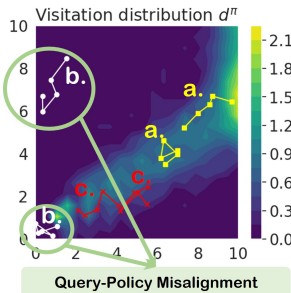

We observe that the selected segments of existing query selection schemes typically fall outside the scope of the visitation distribution $d^\pi$ (marked with green circles). We refer to this phenomenon as *query-policy misalignment*, as illustrated in Figure 3. Such misalignment wastes valuable feedback because the overseer provides preferences on experiences that are less likely encountered by, or in other words, irrelevant to the current RL agent's learning. Therefore, although these selected segments

Figure 3: *Query-policy misalignment.* Existing query selection methods often select queries that lie outside the visitation distribution of the current policy.

may improve the overall quality of the reward model in the full state-action space, they contribute little to current policy training and potentially cause feedback inefficiency. By contrast, *policy-aligned selection* selects fresh segments that are recently visited by the current RL policy, enabling timely feedback on the current status of the policy and leading to significant performance gain, as shown in Figure 2(c). Furthermore, while the learned reward of *policy-aligned query selection* may differ from the ground truth reward, it provides the most useful information to guide the agent to navigate toward the right direction and more strictly discourages detours as compared to the vague per-step ground truth reward. This suggests that the learned reward captures more targeted information in solving the task while also blocking out less useful information in the ground truth reward, thus enabling more effective policy learning. For additional evidence substantiating the existence of the *query-policy misalignment* issue and its contribution to feedback inefficiency, please refer to the Appendix D.1.

As mentioned earlier, existing PbRL methods struggle with feedback inefficiency caused by *query-policy misalignment*. To address this issue, we propose an elegant method: QPA (Query-Policy Alignment). The key idea of QPA is that rather than learning the reward model across the entire global state-action space inadequately as existing methods do, it is more effective to focus on learning the reward model precisely within the distribution of the current policy $d^\pi$, utilizing the same amount of human feedback. On the other hand, the Q-function should also be accurately approximated around $d^\pi$. We present an error bound in Appendix A to intuitively explain why this idea is reasonable.

## 5  QUERY-POLICY ALIGNMENT FOR PREFERENCE-BASED RL (QPA)

In this section, we introduce an elegant solution: QPA, which can effectively address *query-policy misalignment* and is also compatible with existing off-policy PbRL methods with **only 20 lines of code** modifications. Please see Algorithm 1 for the outline of our method.

### 5.1  POLICY-ALIGNED QUERY SELECTION

In contrast to existing query selection schemes, we highlight that the *segment query selection should be aligned with the on-policy distribution*. In particular, it is crucial to ensure that the pairs of segments $(\sigma^0, \sigma^1)$ selected for preference queries stay close to the current policy's visitation distribution $d^\pi$. By assigning more overseer's feedback to segments obtained from on-policy trajectories of the current policy $\pi$, we aim to enhance the accuracy of the preference (reward) predictor within the on-policy distribution $d^\pi$. We refer to this query scheme as *policy-aligned query selection*.

In practice, a natural approach to implement policy-aligned query selection is to utilize the current policy $\pi$ to interact with the environment and generate a set of trajectories. Then, pairs of segments $(\sigma^0, \sigma^1)$ can be selected from these trajectories to obtain overseer's preference query. While such "absolute" policy-aligned query selection ensures that all selected segments conform to the on-policy distribution $d^\pi$, it may have a negative impact on the sample efficiency of off-policy RL due to the additional on-policy rollout. Instead of performing the "absolute" policy-aligned query selection, an alternative is to select segments that are within or "near" on-policy distribution. As we mentioned in Section 3, in typical off-policy PbRL, $(\sigma^0, \sigma^1)$ are selected in trajectories sampled from the RL replay buffer $\mathcal{D}$. A simple yet effective way to perform policy-aligned query selection is to

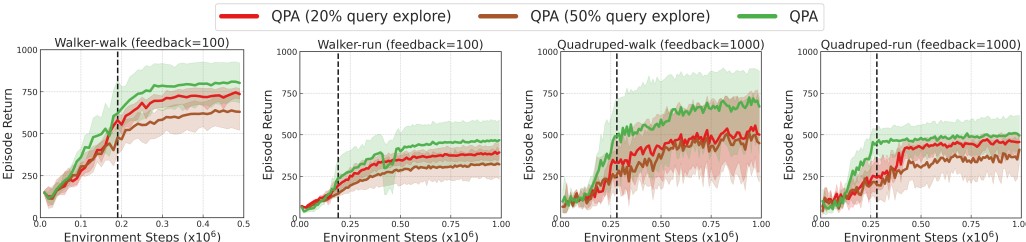

Figure 4: Learning curves of QPA and QPA (20% or 50% query explore) on locomotion tasks. In QPA, 100% of the queries are sampled from the policy-aligned buffer $\mathcal{D}^{\text{pa}}$. In QPA (20% or 50% query explore), we sample 20% or 50% queries from the entire replay buffer $\mathcal{D}$, while the remaining 80% or 50% of queries are sampled from $\mathcal{D}^{\text{pa}}$. The declining performance of QPA (20% or 50% query explore) suggests that allocating some queries to the entire replay buffer in an attempt to explore high-reward state regions can compromise feedback efficiency.

choose $(\sigma^0, \sigma^1)$ from the most recent trajectories stored in $\mathcal{D}$. For the sake of clarity, we refer to the buffer that stores the most recent trajectories as the policy-aligned buffer $\mathcal{D}^{\text{pa}}$. It's worth noting that this simple approach strikes a balance between policy-aligned query and sample efficiency in RL. Furthermore, it is particularly easy to implement and enables a minimalist modification to existing PbRL methods. Take the well-known and publicly-available B-pref (Lee et al., 2021a) PbRL implementation framework as an example, all that's required is to reduce the size of the first-in-first-out query selection buffer, which can be thought of as a knockoff of $\mathcal{D}^{\text{pa}}$.

The use of policy-aligned query confines the query selection to the policy-aligned buffer $\mathcal{D}^{\text{pa}}$, which can avoid the uninformative exploration of query selection within the entire replay buffer $\mathcal{D}$. We show in Figure 4 that the excessive exploration of query selection could hurt feedback efficiency. Querying segments from entire replay buffer $\mathcal{D}$ in an attempt to explore high-reward state regions can only get the high-reward segments *by chance*. Allocating more queries to $\mathcal{D}$ cannot guarantee finding high-reward regions, but potentially resulting in wasted human feedback. In contrast, the queried segments only sampled in $\mathcal{D}^{\text{pa}}$ can provide valuable information for learning the rewards within on-policy distribution, resulting in continual improvement in current policy. Given the limited feedback typically encountered in real-world scenarios, it is preferable to help current policy to get more local guidance using policy-aligned query selection. Please refer to Appendix D.4 for more discussions and experiments about the exploration in PbRL.

## 5.2 Hybrid Experience Replay

Following the policy-aligned query selection and reward learning procedure, it's also important to ensure that *value learning is aligned with the on-policy distribution*. Specifically, more attention should be paid to improving the veracity of Q-function within on-policy distribution $d^\pi$, where the preference (reward) predictor performs well in the preceding step.

To update the Q-function, existing off-policy PbRL algorithms perform the empirical Bellman iteration Eq.(1) by simply sampling transitions $(s, a, s')$ uniformly from the replay buffer $\mathcal{D}$. Although the aforementioned policy-aligned query selection can effectively facilitate the learning of the reward function within $d^\pi$, the Q-function may not be accurately approximated on $d^\pi$ due to inadequate empirical Bellman iterations using $(s, a, s')$ transitions drawn from $d^\pi$. Taking inspiration from combined Q-learning (Zhang & Sutton, 2017) and some prioritized experience replay methods that consider on-policyness (Liu et al., 2021; Sinha et al., 2022), we devise a *hybrid experience replay* mechanism. To elaborate, we still sample transitions $(s, a, s')$ uniformly, but from two different sources. Specifically, half of the uniformly sampled transitions are drawn from $\mathcal{D}$, while the other half are drawn from the policy-aligned buffer $\mathcal{D}^{\text{pa}}$. In other words, the hybrid sample ratio between $\mathcal{D}^{\text{pa}}$ and $\mathcal{D}$ in experience replay is set to 0.5. The proposed mechanism can provide assurance that the Q-function is updated adequately near $d^\pi$.

## 5.3 Data Augmentation for Reward Learning

Besides the above two key designs, we also adopt the temporal data augmentation technique for reward learning (Park et al., 2022). To further clarify, we randomly subsample several shorter pairs of snippets $(\hat{\sigma}^0, \hat{\sigma}^1)$ from the queried segments $(\sigma^0, \sigma^1, y)$, and put these $(\hat{\sigma}^0, \hat{\sigma}^1, y)$ into the preference buffer $\mathcal{D}^\sigma$ for optimizing the cross-entropy loss in Eq.(4). Data augmentation has been widely used in many deep RL (Kostrikov et al., 2020; Laskin et al., 2020a;b) algorithms, and also has been seamlessly integrated into previous PbRL algorithms for consistency regularization (Park et al., 2022). Diverging from the practical implementation in Park et al. (2022), we generate multiple $(\hat{\sigma}^0, \hat{\sigma}^1, y)$

instances from a single $(\sigma^0, \sigma^1, y)$, as opposed to generating only one $(\hat{\sigma}^0, \hat{\sigma}^1, y)$ instance from each $(\sigma^0, \sigma^1, y)$, which effectively expands the preference dataset. An in-depth ablation analysis of the data augmentation technique is provided in Section 6.3 and Appendix D.5.

# 6 EXPERIMENT

In this section, we present extensive evaluations on 6 locomotion tasks in DMControl (Tassa et al., 2018) and 3 robotic manipulation tasks in MetaWorld (Yu et al., 2020).

## 6.1 IMPLEMENTATION AND EXPERIMENT SETUPS

QPA can be incorporated into any off-policy PbRL algorithms. We implement QPA on top of the widely-adopted PbRL backbone framework B-Pref (Lee et al., 2021a). We configure the policy-aligned query selection with a policy-aligned buffer size of $N = 10$. The hybrid experience replay is implemented with a sample ratio of $\omega = 0.5$, and we set the data augmentation ratio at $\tau = 20$.

In our experiments, we demonstrate the efficacy of QPA in comparison to PEBBLE (Lee et al., 2021b), the SOTA method, SURF (Park et al., 2022), and on-policy PbRL method PrePPO (Christiano et al., 2017). As QPA, PEBBLE, and SURF all employ SAC (Haarnoja et al., 2018) for policy learning, we utilize SAC with ground truth reward as a reference performance upper bound for these approaches. For PEBBLE and SURF, we employ the disagreement query selection scheme in their papers that selects segments with the largest ensemble disagreement of reward models (Lee et al., 2021b; Park et al., 2022; Ibarz et al., 2018). To be specific, we train an ensemble of three reward networks $\hat{r}_\psi$ with varying random initializations and select $(\sigma^0, \sigma^1)$ based on the variance of the preference predictor $P_\psi$. While leveraging an ensemble of reward models for query selection may offer improved robustness and efficacy in complex tasks as observed in (Lee et al., 2021b; Ibarz et al., 2018), the additional computational cost incurred by multiple reward models can be unacceptable in scenarios where the reward model is particularly large, *e.g.,* large language models (LLM). Hence in QPA, we opt to use a **single** reward model and employ policy-aligned query selection (simply randomly selects segments from the on-policy buffer $\mathcal{D}^{\mathrm{pa}}$). After all feedback is provided and the reward learning phase is complete, we switch from the hybrid experience replay to the commonly used uniform experience replay for policy evaluation in QPA.

In each task, QPA, SURF, PEBBLE and PrePPO utilize the same amount of total preference queries and feedback frequency for a fair comparison. We use an oracle scripted overseer to provide preferences based on the cumulative ground truth rewards of each segment defined in the benchmarks. Using the scripted ground truth overseer allows us to evaluate the performance of PbRL algorithms quantitatively, unbiasedly and quickly, which is a common practice in previous PbRL literature (Christiano et al., 2017; Lee et al., 2021b;a; Park et al., 2022; Liang et al., 2022). We perform 10 evaluations on locomotion tasks and 100 evaluations on robotic manipulation tasks across 5 runs every $10^4$ environment steps and report the mean (solid line) and 95% confidence interval (shaded regions) of the results, unless otherwise specified. Please see Appendix C.2 for more experimental details.

## 6.2 BENCHMARK TASKS PERFORMANCE

**Locomotion tasks in DMControl suite.** DMControl (Tassa et al., 2018) provides diverse high-dimensional locomotion tasks based on MuJoCo physics (Todorov et al., 2012). We choose 6 complex tasks in DMControl: *Walker_walk, Walker_run, Cheetah_run, Quadruped_walk, Quadruped_run, Humanoid_stand*. Figure 5 shows the learning curves of SAC (green), QPA (red), SURF (brown), and PEBBLE (blue) on these tasks. As illustrated in Figure 5, QPA enjoys significantly better feedback efficiency and outperforms SURF and PEBBLE by a substantial margin on all the tasks. To be mentioned, the SOTA method SURF adopts a pseudo-labeling based semi-supervised learning technique to enhance feedback efficiency. By contrast, QPA removes these complex designs and achieves consistently better performance with a minimalist algorithm. To further evaluate feedback efficiency of QPA, we have included additional experiment results in Appendix D.3, showcasing the performance of these methods under varying total amounts of feedback and different feedback frequencies. Surprisingly, we observe that in some complex tasks (e.g., *Quadruped_walk, Quadruped_run*), QPA can even surpass SAC with ground truth reward during the early training stages, despite experiencing stagnation of performance improvement as feedback provision is halted in the later stages. We provide additional experiment results in Section 6.4 to further elaborate on this phenomenon.

**Robotic manipulation tasks in Meta-world.** We conduct experiments on 3 complex manipulation tasks in Meta-world (Yu et al., 2020): *Door-unlock, Drawer-open, Door-open*. The learning curves

are presented in Figure 6. Similar to prior works (Christiano et al., 2017; Lee et al., 2021b;a; Park et al., 2022; Liang et al., 2022), we employ the ground truth success rate as a metric to quantify the performance of these methods. Once again, these results provide further evidence that QPA effectively enhances feedback efficiency across a diverse range of complex tasks.

The dashed black line depicted in Figure 5, 6 and subsequent figures represents the last feedback collection step. The stopping step for feedback collection is set to align with that of previous work (Park et al., 2022), facilitating a straightforward cross-referencing for readers. We conduct additional experiments with different feedback stop steps in Appendix D.3. While increased feedback enhances the performance of PbRL methods, the limited amount of total feedback enables a more effective evaluation of the feedback efficiency of these PbRL methods. In MetaWorld environment, the variance of these PbRL algorithms increases, which has also been observed in other PbRL literature (Lee et al., 2021b; Park et al., 2022; Liang et al., 2022). For a clear comparison, we provide Table 5 in Appendix D.3 to summarize the mean and standard deviation of these PbRL methods.

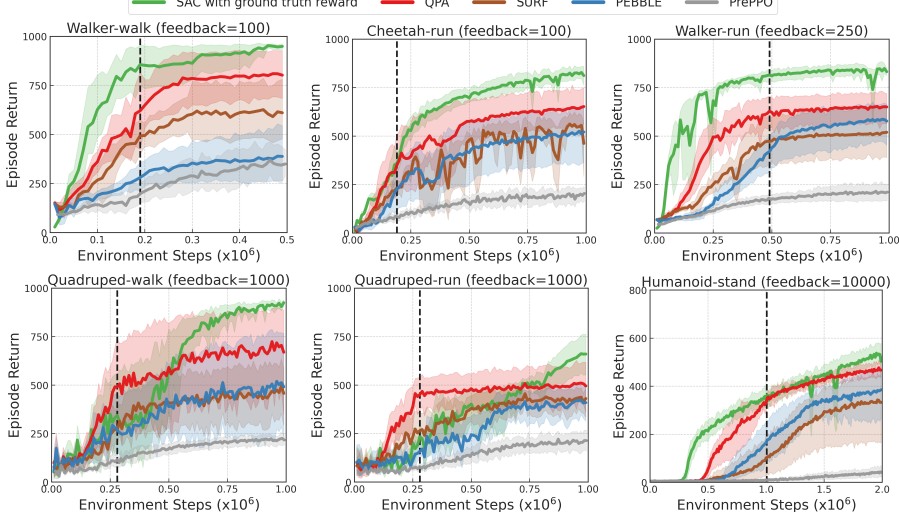

Figure 5: Learning curves on locomotion tasks as measured on the ground truth reward. The dashed black line represents the last feedback collection step.

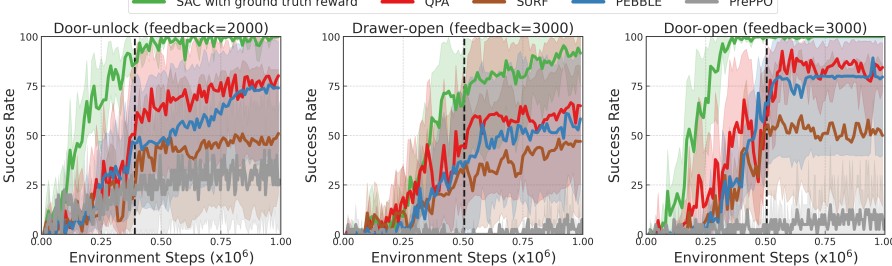

Figure 6: Learning curves on robotic manipulation tasks as measured on the ground truth success rate. The dashed black line represents the last feedback collection step.

### 6.3 ABLATION STUDY

To evaluate the impact of each component in QPA, we incrementally apply policy-aligned query (PA), hybrid experience replay (HR), and data augmentation (DA) to the backbone algorithm PEBBLE. Figure 7 confirms that policy-aligned query has a positive impact on final results. Moreover, the combination of policy-aligned query and experience replay proves to be indispensable for the success of our method. While data augmentation does not always guarantee performance improvement, it tends to enhance performance in most cases. In certain tasks, using only the hybrid experience replay technique can result in slightly improved performance. This could be attributed to the advantage of on-policyness (Liu et al., 2021; Sinha et al., 2022) that is facilitated by hybrid experience replay.

QPA also exhibits good hyperparameter robustness and achieves consistent performance improvement over SURF and PEBBLE across various tasks with different parameter values of the query buffer size $N$, data augmentation ratio $\tau$ and hybrid experience replay sample ratio $\omega$. We provide detailed

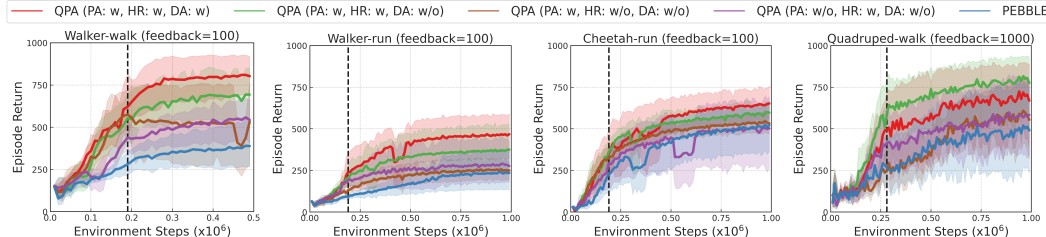

Figure 7: Contribution of each technique in QPA, i.e., policy-aligned query (PA), hybrid experience replay (HR), and data augmentation (DA).

ablation results on hyperparameters in Appendix D.5. In all the tasks presented in Figure 5 and Figure 6, we utilize a data augmentation ratio of $\tau = 20$ and a hybrid experience replay sample ratio of $\omega = 0.5$. For the majority of tasks, we set the size of the policy-aligned buffer as $N = 10$.

### 6.4 ADDITIONAL BENEFIT OF THE REWARD LEARNED BY QPA

As observed in Section 6.2, although SAC with ground truth reward is expected to attain higher performance as compared to PbRL algorithms built upon SAC, it is found that QPA can surpass it during the early training stages in certain tasks. To further investigate this phenomenon, we conduct experiments that increase the total amount of feedback and allow the overseer to provide feedback throughout the training process. As illustrated in Figure 8, QPA often exhibits faster learning compared to SAC with ground truth reward in this setting, and in *Quadruped_run* task even achieves higher scores. This phenomenon may be attributed to the ability to **encode the long-term horizon information** of the reward function learned by QPA, which can be more beneficial for the **current policy** to learn and successfully solve the task. Such a property is also uncovered in the motivating example in Section 4. This intriguing and noteworthy phenomenon also highlights the possibility of PbRL methods with learned rewards outperforming RL methods with per-step ground truth rewards. We hope that this observation will inspire further investigations into the essence of learned rewards in PbRL and foster the development of more feedback-efficient PbRL methods in the future.

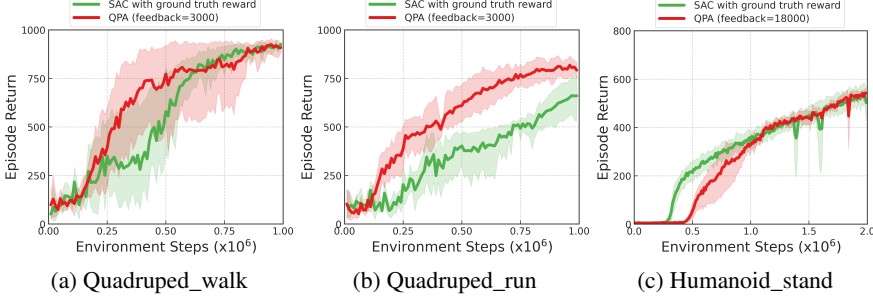

(a) Quadruped_walk    (b) Quadruped_run    (c) Humanoid_stand

Figure 8: Learning curves of QPA compared to SAC with ground truth reward given more overseer feedback throughout the entire training process .

## 7 CONCLUSION AND DISCUSSION

This paper addresses a long-neglected issue in existing PbRL studies, namely *query-policy misalignment*, which hinders the existing query selection schemes from effectively improving feedback efficiency. To tackle this issue, we propose a bidirectional query-policy alignment (QPA) approach that incorporates *policy-aligned query selection* and *hybrid experience replay*. QPA can be implemented with minimal code modifications on existing PbRL algorithms. Simple yet effective, extensive evaluations on DMControl and MetaWorld benchmarks demonstrate substantial gains of QPA in terms of feedback and sample efficiency, highlighting the importance of addressing the *query-policy misalignment* issue in PbRL research. However, note that the *query-policy misalignment* issue is inherently not present in on-policy PbRL methods, as they naturally select on-policy segments to query preferences. These methods, however, suffer from severe sample inefficiency compared to off-policy PbRL methods. In contrast, our QPA approach not only enables sample-efficient off-policy learning, but also achieves high feedback efficiency, presenting a superior solution for the practical implementation of PbRL in real-world scenarios.

ACKNOWLEDGMENTS

This work is supported by the National Key Research and Development Program of China under Grant (2022YFB2502904, 2022YFA1004600), funding from AsiaInfo Technologies and the National Natural Science Foundation of China under Grant (No. 62125304).

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

## A PROOFS

In this section, we present an error bound to intuitively explain why the key idea behind the two techniques in QPA (e.g. policy-aligned query selection and hybrid experience replay ) is reasonable. Note that we do not aim to provide a tighter bound but rather to offer an insightful theoretical interpretation to *query-policy alignment*.

Given the learned reward function $\widehat{r}_\psi$, the current stochastic policy $\pi$ and its state-action visitation distribution $d^\pi$, we denote $Q^\pi_{\widehat{r}_\psi}$ as the Q-function of $\pi$ associated with $\widehat{r}_\psi$ and $\hat{Q}^\pi_{\widehat{r}_\psi}$ as the *estimated* Q-function obtained from the policy evaluation step in Eq. (1), which serves as an approximation of $Q^\pi_{\widehat{r}_\psi}$. $Q^\pi_r$ denotes the Q-function of $\pi$ with true reward $r$. Define the distribution-dependent norms $\|f(x)\|_\mu := \mathbb{E}_{x\sim\mu}\left[|f(x)|\right]$. We have the following error bound:

*Given the two conditions $\|\widehat{r}_\psi - r\|_{d^\pi} \leq \epsilon$ and $\|Q^\pi_{\widehat{r}_\psi} - \hat{Q}^\pi_{\widehat{r}_\psi}\|_{d^\pi} \leq \alpha$, the value approximation error $\|Q^\pi_r - \hat{Q}^\pi_{\widehat{r}_\psi}\|_{d^\pi}$ is upper bounded as:*

$$\|Q^\pi_r - \hat{Q}^\pi_{\widehat{r}_\psi}\|_{d^\pi} \leq \frac{\epsilon}{1-\gamma} + \alpha \tag{5}$$

*Proof.* By repeatedly applying triangle inequality, we have

$$\begin{aligned}
\|Q^\pi_r - \hat{Q}^\pi_{\widehat{r}_\psi}\|_{d^\pi} &= \|Q^\pi_r - Q^\pi_{\widehat{r}_\psi} + Q^\pi_{\widehat{r}_\psi} - \hat{Q}^\pi_{\widehat{r}_\psi}\|_{d^\pi} \\
&\leq \|Q^\pi_r - Q^\pi_{\widehat{r}_\psi}\|_{d^\pi} + \|Q^\pi_{\widehat{r}_\psi} - \hat{Q}^\pi_{\widehat{r}_\psi}\|_{d^\pi} \\
&= \mathbb{E}_{(s,a)\sim d^\pi}\left|Q^\pi_r(s,a) - Q^\pi_{\widehat{r}_\psi}(s,a)\right| + \alpha \\
&= \mathbb{E}_{(s,a)\sim d^\pi}|r(s,a) + \gamma\mathbb{E}_{s'\sim T, a'\sim\pi}Q^\pi_r(s',a') \\
&\qquad - \widehat{r}_\psi(s,a) - \gamma\mathbb{E}_{s'\sim T, a'\sim\pi}Q^\pi_{\widehat{r}_\psi}(s',a')| + \alpha \\
&\leq \epsilon + \gamma\|Q^\pi_r - Q^\pi_{\widehat{r}_\psi}\|_{d^\pi} + \alpha \\
&\leq \epsilon + \gamma\epsilon + \gamma^2\epsilon + ... + \gamma^\infty\epsilon + \alpha \\
&= \frac{\epsilon}{1-\gamma} + \alpha
\end{aligned} \tag{6}$$

$\square$

## B ALGORITHM PROCEDURE

We provide the procedure of QPA in Algorithm 1. QPA is compatible with existing off-policy PbRL methods, with only 20 lines of code modifications on top of the widely-adopted PbRL backbone framework B-Pref (Lee et al., 2021a).

## C EXPERIMENTAL DETAILS

### C.1 2D NAVIGATION EXPERIMENT IN SECTION 4

In this section, we present the detailed task descriptions and implementation setups of the motivating example in Section 4.

**Task description.** As illustrated in Figure 2 (a), we consider a 2D continuous space with $(x, y)$ coordinates defined on $[-10, 10]^2$. For each step, the RL agent can move $\Delta x$ and $\Delta y$ ranging from $[-1, 1]$. The objective for the agent is to navigate from the starting point $(1, 1)$ to the goal location $(10, 10)$ as quickly as possible. The hand-engineered reward function (ground truth reward in Figure 2 (b)) to provide preferences is defined as the negative distance to the goal, *i.e.*, $r(s, a) = -\sqrt{(x-10)^2 + (y-10)^2}$.

---

**Algorithm 1:** QPA

---

**Input** : Frequency of overseer feedback $K$, number of queries per feedback session $M$
Policy-aligned buffer size $N$, data augmentation ratio $\tau$, hybrid sample ratio $\omega$
**Initialize** : Initialize replay buffer $\mathcal{D}$, query buffer $\mathcal{D}^\sigma$, policy-aligned buffer $\mathcal{D}^{\mathrm{pa}}$ with size $N$

1    **(Option)** Unsupervised pretraining (Lee et al., 2021b)
2    **for** *each iteration* **do**
3       Collect and store new experience $\mathcal{D} \leftarrow \mathcal{D} \cup \{(s,a,r,s')\}, \mathcal{D}^{\mathrm{on}} \leftarrow \mathcal{D}^{\mathrm{on}} \cup \{(s,a,r,s')\}$
4       **if** *iteration* $\% K == 0$ **then**
         /* policy-aligned query selection (see Section 5.1)        */
5          $\{(\sigma^0, \sigma^1)\}_{i=1}^M \sim \mathcal{D}^{\mathrm{on}}$
6          Query for preferences $\{y\}_{i=1}^M$, and store preference $\mathcal{D}^\sigma \leftarrow \mathcal{D}^\sigma \cup \{(\sigma^0, \sigma^1, y)\}_{i=1}^M$
7          **for** *each gradient step* **do**
8              Sample a minibatch preferences $\mathcal{B} \leftarrow \{(\sigma^0, \sigma^1, y)\}_{i=1}^h \sim \mathcal{D}^\sigma$
             /* Data augmentation for reward learning (see Section 5.3)    */
9              Generate augmented preferences $\hat{\mathcal{B}} \leftarrow \{(\hat{\sigma}^0, \hat{\sigma}^1, y)\}_{i=1}^{h \times \tau}$ based on $\mathcal{B}$
10              Optimize $\mathcal{L}^{\mathrm{reward}}$ in Eq. (4) *w.r.t.* $\widehat{r}_\psi$ using $\hat{\mathcal{B}}$
11          Relabel the rewards (Lee et al., 2021b) in $\mathcal{D}$
12       **for** *each gradient step* **do**
         /* Hybrid experience replay (see Section 5.2)        */
13          Sample minibatch $\mathcal{D}_{\mathrm{mini}} \leftarrow \{(s,a,r,s')\}_{i=1}^{\frac{n}{2}} \sim \mathcal{D}, \mathcal{D}^{\mathrm{on}}_{\mathrm{mini}} \leftarrow \{(s,a,r,s')\}_{i=1}^{\frac{n}{2}} \sim \mathcal{D}^{\mathrm{on}}$
14          Optimize SAC agent using $\mathcal{D}_{\mathrm{mini}} \cup \mathcal{D}^{\mathrm{on}}_{\mathrm{mini}}$

---

**Implementation details.** We train PEBBLE (Lee et al., 2021b) using 3 different query selection schemes: *uniform query selection*, *disagreement query selection*, and *policy-aligned query selection* (see Section 5.1). In each feedback session, we can obtain one pair of segments to query overseer preferences. The total amount of feedback is set to 8. Each segment contains 5 transition steps. For all schemes, we select the 1st queries using *uniform query selection* according to PEBBLE implementation. After the 1st query selection, we start selecting queries using different selection schemes. The 2nd selected pairs of segments of different schemes are tracked in Figure 3, and the 3rd to 5th selected pairs are tracked in Figure 2 (d).

## C.2   DMCONTROL AND META-WORLD EXPERIMENTS

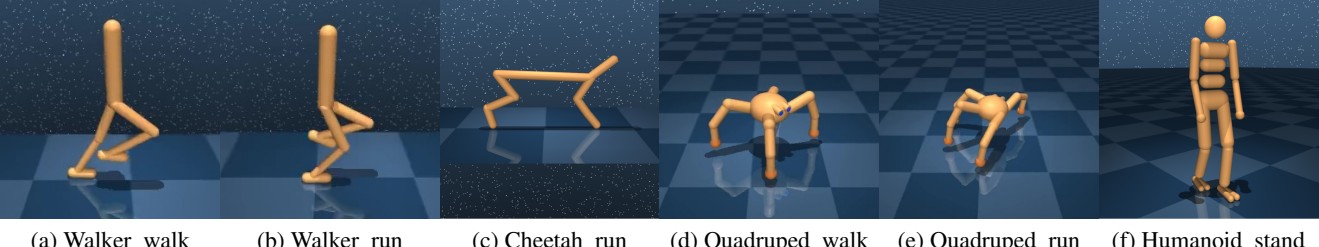

(a) Walker_walk     (b) Walker_run     (c) Cheetah_run     (d) Quadruped_walk     (e) Quadruped_run     (f) Humanoid_stand

Figure 9: Rendered images of locomotion tasks from DMControl.

### C.2.1   TASK DESCRIPTIONS

**Locomotion tasks in DMControl suite.**    DMControl (Tassa et al., 2018) provides diverse high-dimensional locomotion tasks. For our study, we choose 6 complex tasks *Walker_walk, Walker_run, Cheetah_run, Quadruped_walk, Quadruped_run, Humanoid_stand* as depicted in Figure 9. In *Walker_walk* and *Walk_run*, the ground truth reward is a combination of terms encouraging an upright torso and forward velocity. The observation space is 24 dimensional, and the action space is 6 dimensional. In *Cheetah_run*, the ground truth reward is linearly proportional to the forward velocity up to a maximum of 10m/s. The observation space is 17 dimensional, and the action space is 6

dimensional. In *Quadruped_walk* and *Quadruped_run*, the ground truth reward includes the terms encouraging an upright torso and forward velocity. The observation space is 58 dimensional, and the action space is 12 dimensional. In *Humanoid_stand*, the ground truth reward is composed of terms that encourage an upright torso, a high head height, and minimal control. The observation space is 67 dimensional, and the action space is 21 dimensional.

**Robotic manipulation tasks in Meta-world.** Meta-world (Yu et al., 2020) provides diverse high-dimensional robotic manipulation tasks. For all Meta-world tasks, the observation space is 39 dimensional and the action space is 4 dimensional. For our study, we choose 3 complex tasks *Door_unlock, Drawer_open, Door_open* as depicted in Figure 9. For *Door_unlock*, the goal is to unlock the door by rotating the lock counter-clockwise and the initial door position is randomized. For *Drawer_open*, the goal is to open a drawer and the initial drawer position is randomized. For *Door_open*, the goal is to open a door with a revolving joint and the initial door position is randomized. Please refer to (Yu et al., 2020) for detailed descriptions of the ground truth rewards.

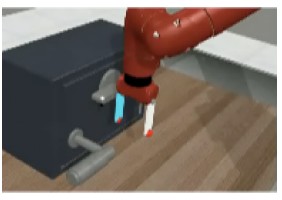 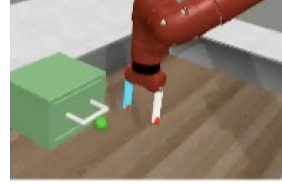 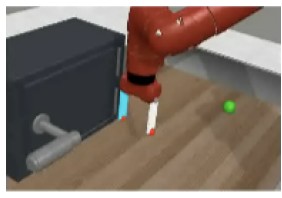

(a) Door_unlock         (b) Drawer_open         (c) Door_open

Figure 10: Rendered images of robotic manipulation tasks from Meta-world.

### C.2.2 IMPLEMENTATION DETAILS

**Implementation framework.** We implement QPA on top of the widely-adopted PbRL backbone framework B-Pref[1] (Lee et al., 2021a). B-Pref provides a standardized implementation of PEBBLE (Lee et al., 2021b), which can be regarded as the fundamental backbone algorithm for off-policy actor-critic PbRL algorithms. Therefore, similar to prior works (SURF (Park et al., 2022), RUNE (Liang et al., 2022), etc), we opt for PEBBLE as our backbone algorithm as well. B-Pref implements 3 distinct buffers: the RL replay buffer $\mathcal{D}$, the query selection buffer $\mathcal{D}'$ and the preference buffer $\mathcal{D}^\sigma$. $\mathcal{D}$ stores historical trajectories for off-policy RL agent's training. $\mathcal{D}'$ is a copy of $\mathcal{D}$ excluding the predicted reward. It is specifically utilized for segment query selection. $\mathcal{D}^\sigma$ stores the historical feedback $(\sigma^0, \sigma^1, y)$ for reward training. Every time the reward model $\widehat{r}_\psi$ is updated, all of the past experience stored in $\mathcal{D}$ is relabeled accordingly. We implement SURF using their officially released code[2], which is also built upon B-Pref.

**Query selection scheme.** For PEBBLE and SURF, we employ the ensemble disagreement query selection scheme in their papers. To be specific, we train an ensemble of three reward networks $\widehat{r}_\psi$ with varying random initializations. When selecting segments from $\mathcal{D}'$, we firstly uniformly sample a large $\{(\sigma^0, \sigma^1)\}$ batch from $\mathcal{D}'$. Subsequently, we choose $(\sigma^0, \sigma^1)$ from this batch based on the highest variance of the preference predictor $P_\psi$. For QPA, we simply reduce the size of $\mathcal{D}'$. This minimal modification ensures that the first-in-first-out buffer $\mathcal{D}'$ exclusively stores the most recent trajectories, effectively emulating the characteristics of the policy-aligned buffer $\mathcal{D}^{\text{pa}}$. In QPA, instead of utilizing ensemble disagreement query based on multiple reward models, we use a **single** reward model and employ policy-aligned query selection (simply randomly selects segments from the policy-aligned buffer $\mathcal{D}^{\text{pa}}$). This approach significantly reduces the computational cost compared to ensemble disagreement query selection, particularly in scenarios where the reward model is notably large, such as large language models (LLMs). Although leveraging an ensemble of reward models for query selection may offer improved robustness and efficacy in complex tasks as observed in (Lee et al., 2021b; Ibarz et al., 2018), we showcase that QPA, employing the simple policy-aligned query selection, can substantially outperform SURF and PEBBLE with ensemble disagreement query selection.

---

[1] https://github.com/rll-research/BPref
[2] https://openreview.net/forum?id=TfhfZLQ2EJO

**Data augmentation.** In the officially released code of SURF, they implement the temporal data augmentation, which generates one $(\hat{\sigma}^0, \hat{\sigma}^1, y)$ instance from each $(\sigma^0, \sigma^1, y)$ pair. In contrast to their implementation, in QPA, we generate multiple $(\hat{\sigma}^0, \hat{\sigma}^1, y)$ instances from a single $(\sigma^0, \sigma^1, y)$ pair, which effectively expands the preference dataset. Specifically, we sample multiple pairs of snippets $(\hat{\sigma}^0, \hat{\sigma}^1)$ from the queried segments $(\sigma^0, \sigma^1, y)$. These pairs of snippets $(\hat{\sigma}^0, \hat{\sigma}^1)$ consist of sequences of observations and actions, but they are shorter than the corresponding segments $(\sigma^0, \sigma^1)$. In each pair of snippets $(\hat{\sigma}^0, \hat{\sigma}^1)$, $\hat{\sigma}^0$ has the same length as $\hat{\sigma}^1$, but they have different initial states.

**Hyperparameter setting.** QPA, PEBBLE, and SURF all employ SAC (Soft Actor-Critic) (Haarnoja et al., 2018) for policy learning and share the same hyperparameters of SAC. We provide the full list of hyperparameters of SAC in Table 1. Both QPA and SURF utilize PEBBLE as the off-policy PbRL backbone algorithm and share the same hyperparameters of PEBBLE as listed in Table 2. The additional hyperparameters of SURF based on PEBBLE are set according to their paper and are listed in Table 3. The additional hyperparameters of QPA are presented in Table 4.

Table 1: Hyperparameters of SAC

| Hyperparameter | Value | Hyperparameter | Value |
|---|---|---|---|
| Discount | 0.99 | Critic target update freq | 2 |
| Init temperature | 0.1 | Critic EMA | 0.005 |
| Alpha learning rate | 1e-4 | Actor learning rate | 5e-4 (Walker_walk, |
| Critic learning rate | 5e-4 (Walker_walk, | | Cheetah_run, |
| | Cheetah_run, | | Walker_run) |
| | Walker_run) | | 1e-4 (Other tasks) |
| | 1e-4 (Other tasks) | Actor hidden dim | 1024 |
| Critic hidden dim | 1024 | Actor hidden layers | 2 |
| Critic hidden layers | 2 | Actor activation function | ReLU |
| Critic activation function | ReLU | Optimizer | Adam |
| Bacth size | 1024 | | |

Table 2: Hyperparameters of PEBBLE

| Hyperparameter | Value |
|---|---|
| Length of segment | 50 |
| Unsupervised pre-training steps | 9000 |
| Total feedback | 100 (Walker_walk, Cheetah_run, Walker_run) |
| | 1000 (Quadruped_walk, Quadruped_walk) |
| | 2000 (Door_unlock) |
| | 3000 (Drawer_open, Door_open) |
| | 10000 (Humanoid_stand) |
| Frequency of feedback | 5000 (Humanoid_stand, Drawer_open, Door_open) |
| | 20000 (Walker_walk, Cheetah_run, Walker_run, Door_unlock) |
| | 30000 (Quadruped_walk, Quadruped_walk) |
| # of queries per session | 10 (Walker_walk, Cheetah_run, Walker_run) |
| | 30 (Drawer_open, Door_open) |
| | 50 (Humanoid_stand) |
| | 100 (Quadruped_walk, Quadruped_walk, Door_unlock) |
| Size of query selection buffer | 100 |

Table 3: Additional hyperparameters of SURF

| Hyperparameter | Value |
| --- | --- |
| Unlabeled batch ratio | 4 |
| Threshold | 0.999 (Cheetah_run), 0.99 (Other tasks) |
| Loss weight | 1 |
| Min/Max length of cropped segment | [45, 55] |
| Segment length before cropping | 60 |

Table 4: Additional hyperparameters of QPA

| Hyperparameter | Value |
| --- | --- |
| Size of policy-aligned buffer $N$ | 30 (Drawer_open, Door_unlock), 60 (Door_open), 10 (Other tasks) |
| Data augmentation ratio $\tau$ | 20 |
| Hybrid experience replay sample ratio $\omega$ | 0.5 |
| Min/Max length of subsampled snippets | [35, 45] |

## D  MORE EXPERIMENTAL RESULTS

In this section, we present more experimental results. For each task, we perform 10 evaluations across 5 runs every $10^4$ environment steps and report the mean (solid line) and 95% confidence interval (shaded regions) of the results, unless otherwise specified.

### D.1  QUERY-POLICY MISALIGNMENT AND ITS ROLE IN CAUSING FEEDBACK INEFFICIENCY.

In section 4, we present a motivating example to clarify the *query-policy misalignment* issue in existing PbRL methods. In this section, we provide more experimental results in complex environment to show that:

- the query-policy misalignment issue does exist in typical PbRL methods and does cause the feedback inefficiency;
- using only policy-aligend query selection to address the query-policy misalignment can result in a significant performance improvement.

As all the recent off-policy PbRL methods are built upon PEBBLE, we compare the two method: PEBBLE and PEBBLE + policy-aligned query selection. The policy-aligned query selection is presented in Section 5.1. To assess the extent to which the queried segments align with the distribution of the current policy $\pi$, we compute the log likelihood of $\pi$ using the queried segments at each query time. Figure 11 (a) shows that the segments queried by PEBBLE exhibit a low log likelihood of $\pi$, indicating that these segments fall outside the distribution of the current policy $\pi$. Figure 11 (a) and (b) demonstrate that by only using the policy-aligend query selection to address the query-policy misalignment leads to a considerable improvement in feedback efficiency. As the technique policy-aligned query selection can be directly added to PEBBLE without other modifications to address query-policy misalignment and improve feedback efficiency, the query-policy misalignment issue is indeed the key factor causing feedback inefficiency.

### D.2  ADDITIONAL COMPARISON

In this section, we demonstrate the efficacy of QPA in comparison to another SOTA method MRN (Liu et al., 2022). All the basic settings remain consistent with the experimental setup outlined in Section 6.1. The additional hyperparameters (such as the bi-level update frequency) of MRN based on PEBBLE are set according to their paper. Figure 12 shows that in most tasks, QPA consistently outperforms MRN. It is worth noting that MRN adopts the performance of the Q-function as the learning target to formulate a bi-level optimization problem, which is orthogonal to our methods.

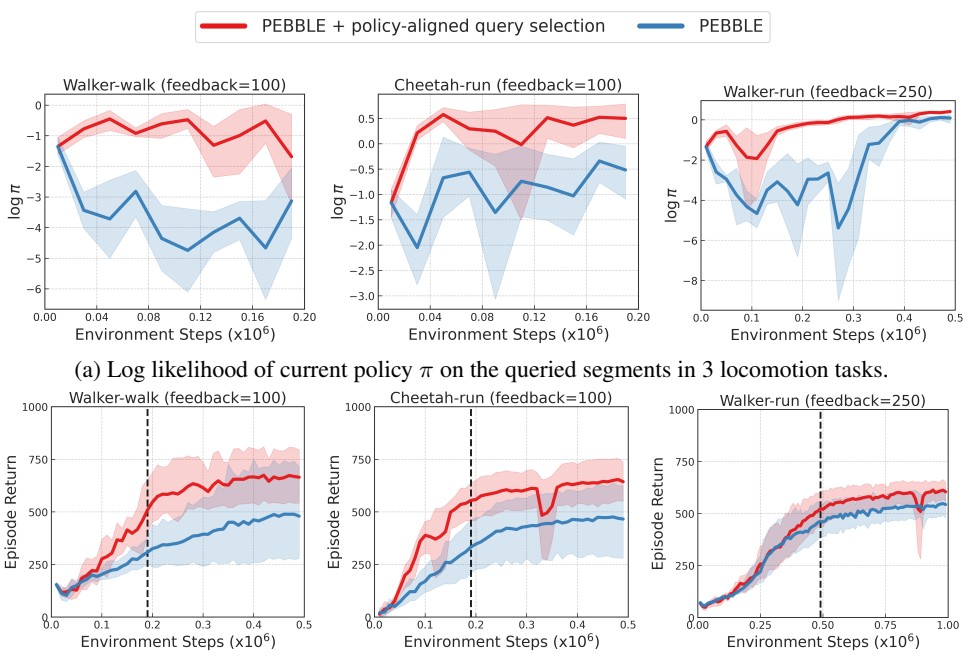

(a) Log likelihood of current policy $\pi$ on the queried segments in 3 locomotion tasks.

(b) Learning curves of episode return in above locomotion tasks.

Figure 11: The blue line represents the typical PbRL method: PEBBLE, upon which most recent PbRL methods are built. The red line represents PEBBLE + policy-aligned query selection. The experiments are conducted on *Walker_walk, Cheetah_run, Walker_run*. **(a)** We use the queried segments at every query time to compute the log likelihood of current policy $\pi$. The segments queried by PEBBLE exhibit a low log likelihood of $\pi$, indicating that these segments fall outside the distribution of the current policy $\pi$, which supports the existence of *query-policy misalignment*. When PEBBLE incorporates our proposed technique *policy-aligned query selection* in Section 5.1, there is a substantial increase in the log likelihood of $\pi$. **(b)** The performance of PEBBLE + policy-aligned query selection significantly surpasses that of PEBBLE in the corresponding task.

Policy-alignment query selection can be easily incorporate into MRN to further improve the feedback efficiency.

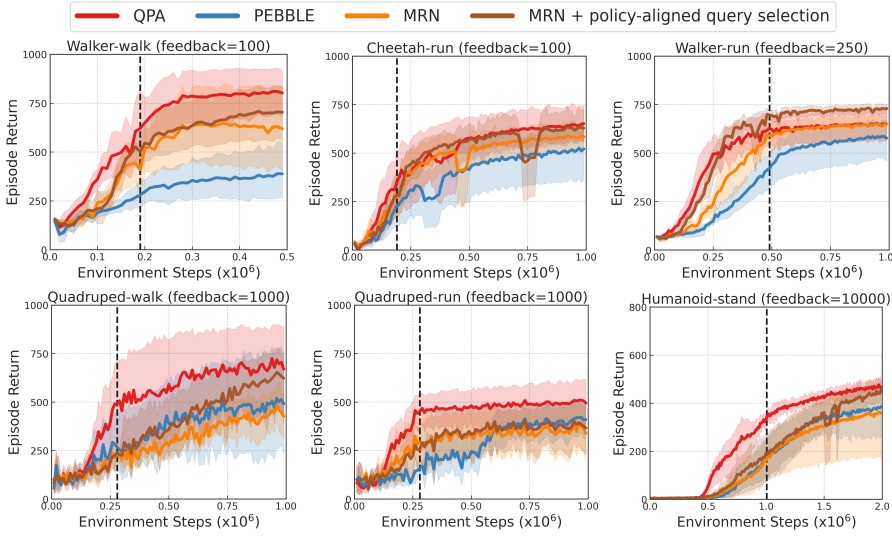

Figure 12: Learning curves on locomotion tasks of QPA, PEBBLE, MRN, and MRN + policy-aligned query selection. The dashed black line represents the last feedback collection step.

### D.3 RESULTS UNDER VARYING TOTAL FEEDBACK AND FEEDBACK FREQUENCIES

To further demonstrate the feedback efficiency of QPA, we compare its performance with that of SURF and PEBBLE under varying total feedback and different feedback frequencies on certain locomotion tasks. As illustrated in Figure 13-17, QPA (red) consistently outperforms SURF (brown) and PEBBLE (blue) across a wide range of total feedback quantities and feedback frequencies. The dashed black line depicted in these figures represents the last feedback collection step.

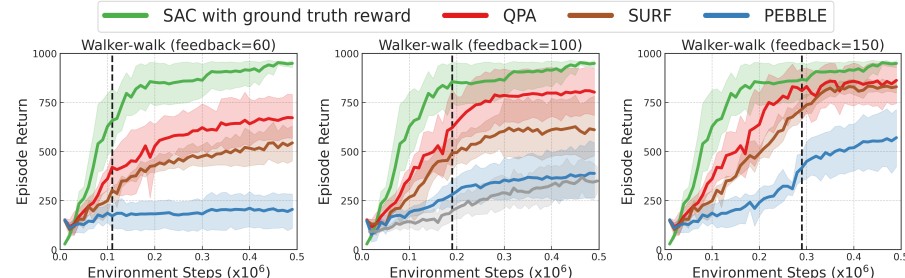

Figure 13: Learning curves on *Walker_walk* under varying total amounts of feedback {60, 100, 150}.

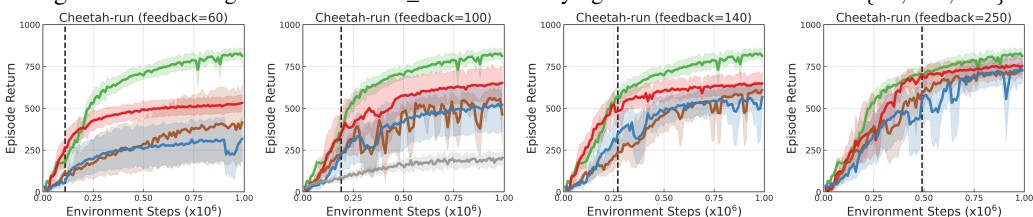

Figure 14: Learning curves on *Cheetah_run* under varying total amounts of feedback {60, 100, 140, 250}.

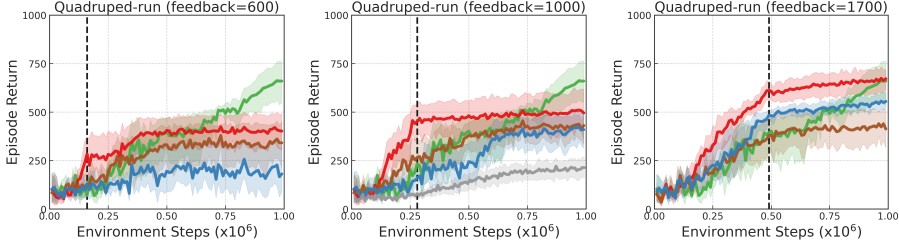

Figure 15: Learning curves on *Quadruped_run* under varying total amounts of feedback {600, 1000, 1700}.

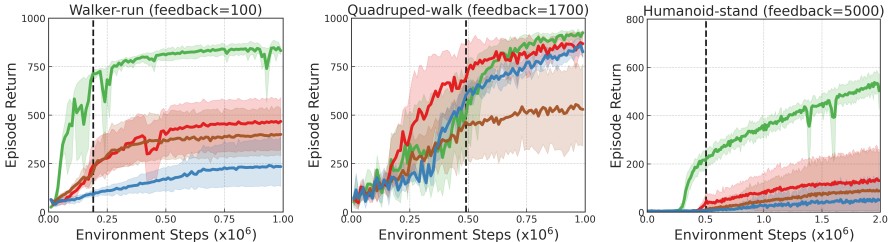

Figure 16: Learning curves on other tasks under different total amounts of feedback.

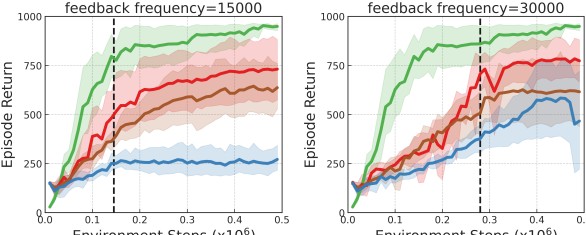

Figure 17: Learning curves on *Walker_walk* under varying feedback frequencies.

Additionally, we provide a summary in Table 5 showcasing the average episode return or success rate of all PbRL methods across all domains, including standard deviation, based on the final 10 or 100 evaluations and 5 seeds.

Table 5: Average episode return / success rate of all PbRL methods with standard deviation over final 10/100 evaluations and 5 seeds. QPA consistently outperforms other methods across a wide range of tasks and given feedback amounts.

| Task - Total_feedback | SAC with ground truth reward | PEBBLE | SURF | QPA |
|---|---|---|---|---|
| Walker_walk-100 | 949.21±21.94 | 388.58±163.00 | 610.93±170.83 | **802.65±133.44** |
| Walker_walk-60 | 949.21±21.94 | 205.23±103.08 | 544.06±105.24 | **672.19±128.97** |
| Walker_walk-150 | 949.21±21.94 | 570.68±168.21 | 829.95±38.05 | **862.39±127.86** |
| Walker_run-100 | 832.44±21.91 | 234.48±147.47 | 400.4±144.47 | **467.63±152.58** |
| Cheetah_run-100 | 812.35±60.47 | 521.44±162.35 | 462.44±167.73 | **652.56±109.09** |
| Cheetah_run-60 | 812.35±60.47 | 301.13±144.21 | 378.76±255.74 | **528.91±59.16** |
| Cheetah_run-140 | 812.35±60.47 | 499.01±203.62 | 590.98±61.87 | **647.55±64.62** |
| Quadruped_walk-1000 | 925.25±20.09 | 491.04±301.0 | 457.56±168.97 | **669.75±301.61** |
| Quadruped_run-1000 | 660.44±128.7 | 409.87±113.58 | 431.03±94.44 | **495.71±139.24** |
| Quadruped_run-600 | 660.44±128.7 | 166.93±148.51 | 342.07±103.45 | **403.95±99.82** |
| Humanoid_stand-10000 | 503.13±85.85 | 385.26±123.9 | 332.67±166.18 | **467.09±32.41** |
| Humanoid_stand-5000 | 503.13±85.85 | 28.11±44.11 | 44.75±77.99 | **79.39±88.63** |
| Door-unlock-2000 | 100.0±0.0 | 74.0±37.65 | 51.0±38.07 | **80.2±37.63** |
| Drawer-open-3000 | 91.6±16.8 | 58.4±40.96 | 47.0±43.41 | **65.0±43.25** |
| Door-open-3000 | 100.0±0.0 | 79.6±39.8 | 50.0±41.91 | **84.4±14.5** |

## D.4    THE EXPLORATION IN QPA

As detailed in Figure 4 and Section 5.1, the exploration of query selection in PbRL can cause feedback inefficiency. The experiments in Section 6.2 and Appendix D.3 also demonstrate that, in general, policy-aligned query selection is more feedback-efficient compared to methods such as uniform query or disagreement query, which, to some extent, "encourage" the exploration of query selection.

Using the policy-aligned query selection, which confines the queries to the local policy-aligned buffer $\mathcal{D}^{pa}$, provides more local guidance for current policy, enhancing feedback efficiency. However, adopting policy-aligned query selection doesn't imply a lack of explorative capability in QPA. In fact, SAC's intrinsic encouragement of exploration can enhance the policy performance of QPA.

We would like to highlight that the exploration of state-action space in traditional RL algorithms and the exploration of query selection in PbRL algorithms are two separate and distinct aspects. While the exploration of query selection potentially causes feedback inefficiency, the exploration ability in RL algorithms can help improve the sample efficiency. QPA adopts SAC as the backbone RL algorithm. The MaxEntropy policy of SAC helps QPA *explore* the unknown regions, while the policy-aligned query helps QPA *exploit* the limited human feedback. We conduct supplementary experiments in Figure 18 that remove the MaxEntropy of SAC. The results show that the intrinsic encouragement of exploration in SAC plays an important role in policy performance.

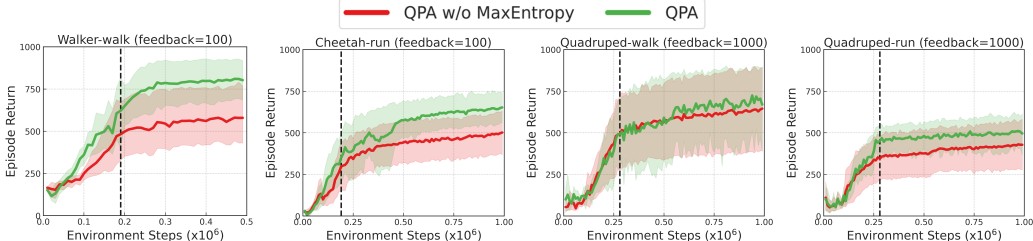

Figure 18: Learning curves of QPA and the modified QPA that removes the MaxEntropy of SAC (QPA w/o MaxEntropy).

## D.5    ADDITIONAL ABLATION STUDY

We emphasize that the outstanding performance of QPA, as demonstrated in Section 6, was **not** achieved by meticulously selecting QPA's hyperparameters. On the contrary, as indicated in Table 4, we consistently use a data augmentation ratio of $\tau = 20$, a hybrid experience replay of $\omega = 0.5$ across all tasks and use a policy-aligned buffer size of $N = 10$ in the majority of tasks. To delve deeper into the impact of hyperparameters on QPA's performance, we conduct an extensive ablation study across a range of tasks.

**Effect of data augmentation ratio $\tau$.** The data augmentation ratio, denoted as $\tau$, represents the number of instances $(\hat{\sigma}^0, \hat{\sigma}^1, y)$ generated from a single pair $(\sigma^0, \sigma^1, y)$. To explore the impact of the data augmentation ratio on QPA's performance, we evaluate the performance of QPA under different data augmentation ratios $\tau \in \{0, 10, 20, 100\}$. Figure 19 illustrates that QPA consistently demonstrates superior performance across a diverse range of data augmentation ratios. While a larger data augmentation ratio does not necessarily guarantee improved performance, it is worth noting that $\tau = 20$ is generally a favorable choice in most cases.

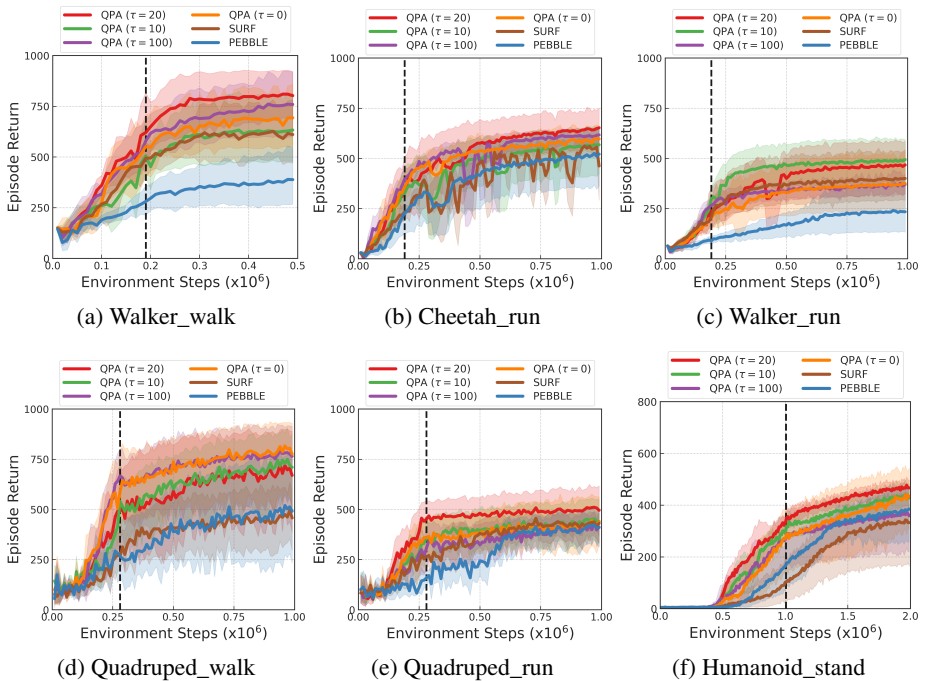

Figure 19: Learning curves on locomotion tasks under different data augmentation ratios $\tau \in \{0, 10, 20, 100\}$ of QPA.

**Effect of policy-aligned buffer size $N$.** The size of the on-policy buffer, denoted as $N$, signifies the number of the recent trajectories stored in the first-in-first-out buffer $\mathcal{D}^{\text{pa}}$. A larger $N$ implies that the policy-aligned buffer $\mathcal{D}^{\text{pa}}$ contain additional historical trajectories generated by a past policy $\pi'$ that may be significantly different from the current policy $\pi$, potentially deviating from the main idea of *policy-aligned query selection*. Therefore, it is reasonable for QPA to opt for a smaller value of $N$. To further investigate how the on-policy buffer size affects QPA's performance, we evaluate the performance of QPA under different on-policy buffer sizes $N \in \{5, 10, 50\}$. Figure 20 demonstrates QPA consistently showcases superior performance, particularly when using a smaller value for $N$.

In *Quadruped_run*, it is observed that QPA with $N = 50$ does not exhibit improved performance. This outcome can be attributed to the fact that the larger value of $N$ compromises the essence of near "on-policy" within $\mathcal{D}^{\text{pa}}$. This once again highlights the significance of the *policy-aligned query selection* principle. A smaller value of $N$ may not always result in a significant improvement in performance. This could be because the pair of segments selected from a very small policy-aligned buffer is more likely to be similar to each other, resulting in less informative queries. Overall, it is generally considered favorable to select $N = 10$ for achieving better performance.

**Effect of hybrid experience replay sample ratio $\omega$.** The sample ratio of hybrid experience replay, denoted as $\omega$, signifies the proportion of transitions sourced from the policy-aligned buffer $\mathcal{D}^{\text{pa}}$ during the policy evaluation (empirical Bellman iteration) step. When $\omega = 0$, all transitions are sourced from the entire replay buffer $\mathcal{D}$. Conversely, when $\omega = 1$, all transitions are sourced from the policy-aligned buffer $\mathcal{D}^{\text{pa}}$. We provide an additional ablation study in Figure 21 to investigate the effect of $\omega$. Only sampling the transitions from the entire replay buffer $\mathcal{D}$ ($\omega = 0.8$) for Bellman iterations diminishes QPA's performance, which validates that the *hybrid experience replay* technique

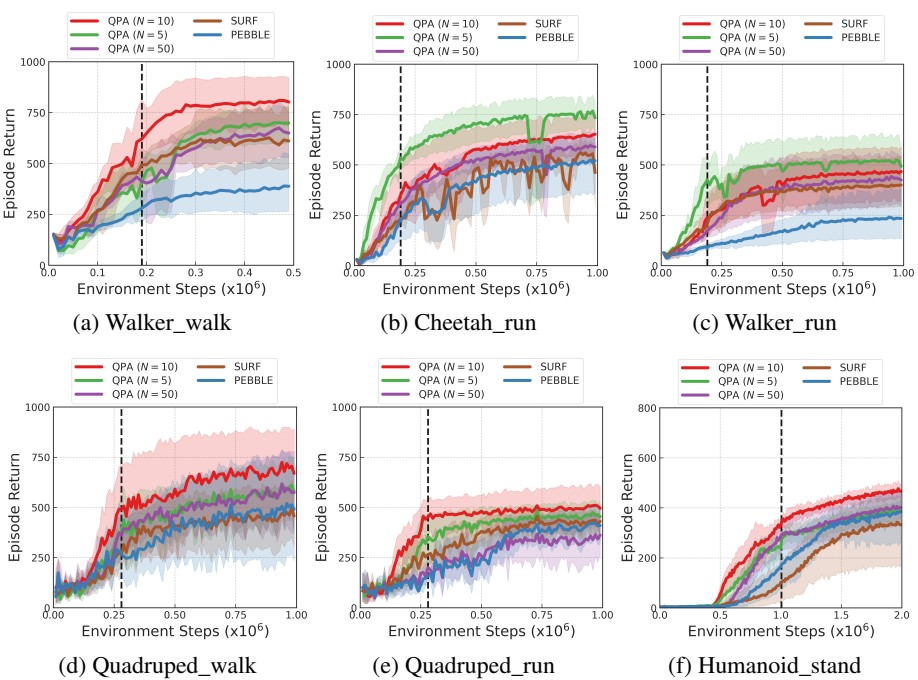

Figure 20: Learning curves on locomotion tasks under different sizes of policy-aligned buffer $N \in \{5, 10, 50\}$ of QPA.

can enhance performance. When a larger proportion of transitions is sampled from the policy-aligned buffer ($\omega = 0.8$), QPA's performance remains close to QPA with $\omega = 0.5$ The *hybrid experience replay* not only benefits the value learning within on-policy distribution, but also improves sample efficiency compared to on-policy PbRL algorithms. Overall, $\omega = 0.5$ is generally a favorable choice.

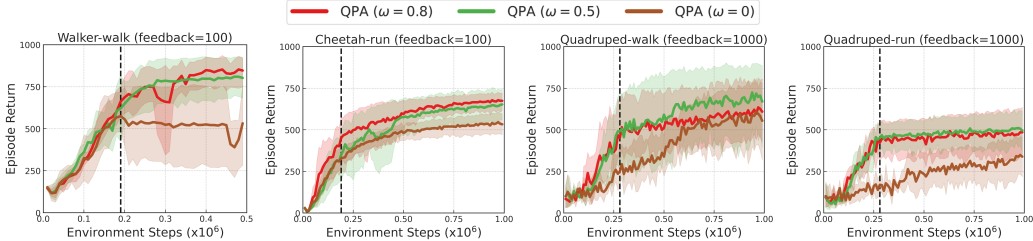

Figure 21: Learning curves on locomotion tasks under different hybrid experience replay sample ratio $\omega \in \{0, 0.5, 0.8\}$.

## E  HUMAN EXPERIMENTS

In this section, we compare the agent trained with real human preferences (provided by the authors) to the agent trained with hand-engineered preferences (provided by the hand-engineered reward function in Appendix C.2.1) on the *Cheetah_run* task. We report the training results in Figure 22. Please refer to the supplementary material for the videos of agent training processes.

**Avoiding reward exploitation via human preferences.** As shown in Figure 22, the agent trained with real human preferences exhibits more natural behavior, while the agent trained with hand-engineered preferences often behaves more aggressively and may even roll over. This is because the hand-engineered reward function is based solely on the linear proportion of forward velocity, without fully considering the agent's posture. Consequently, it can be easily exploited by the RL agent. Take Figure 23 as an example: when comparing the behaviors of "Stand still" and "Recline",

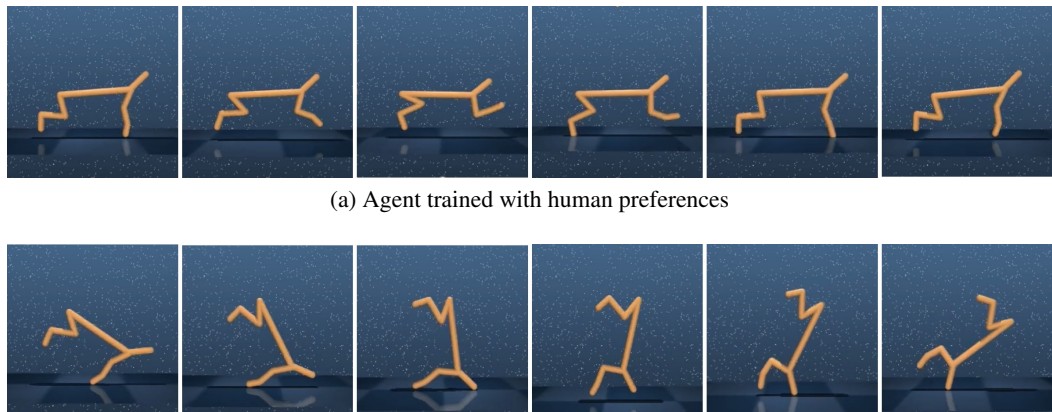

(a) Agent trained with human preferences

(b) Agent trained with hand-engineered preferences

Figure 22: Human experiments on *Cheetah_run* task.

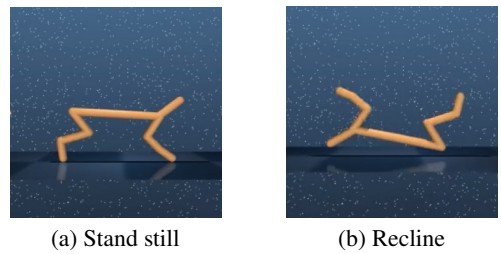

(a) Stand still       (b) Recline

Figure 23: A pair of segments.

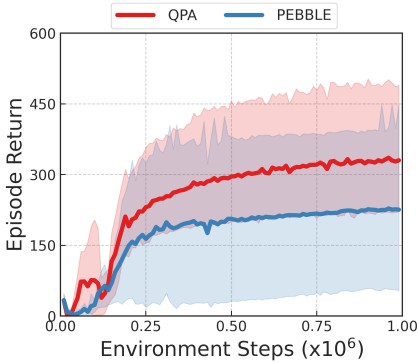

Figure 24: Learning curves of QPA and PEBBLE on *Cheetah_run* under 100 real human preferences.

humans would typically not prefer the latter, as it clearly contradicts the desired behavior of "running forward". However, in our experiments, we observe that the hand-engineered overseer would favor "Recline" over "Stand still", as the hand-engineered preferences only consider the forward velocity and overlook the fact that the agent is lying down while still maintaining some forward velocity. This highlights the advantages of RL with real human feedback over standard RL that training with hand-engineered rewards.

**The effectiveness of QPA under real human preference.**  We also provide the learning curve of QPA compared to PEBBLE under 100 real human preferences in Figure 24. The preferences are labeled by two different humans, which might introduce inconsistencies and contradictions in preferences. Figure 24 shows that using feedback from real humans, our method also notably improves feedback efficiency compared to PEBBLE.

**Experimental details and user interface.**    The total feedback, frequency of feedback and number of queries per session in real human experiments remain consistent with the experimental setup outlined in Section 6.1 and Table 2. A subtle distinction is that human users have the choice to skip certain queries when they find it challenging to determine their preferred segment, unlike the scripted annotator. These skipped queries are not accounted for in the total feedback.

Our implementation of real human PbRL builds upon the widely-adopted PbRL backbone framework B-Pref (Lee et al., 2021a). In the original B-Pref code, scripted preferences based on the cumulative ground truth rewards of each segment defined in the benchmarks are used after each query selection. However, to enable real humans to provide preferences and furnish labeling instructions and a user interface, we've mainly modified the *get_label* function. The modified function now accepts additional inputs: *physics_seg* and *video_recorder*. The *physics_seg* is obtained from *env.physics._physics_state_items()* in DMControl environments (Tassa et al., 2018). *video_recorder* is a class for physics rendering. We provide the Python code of modified *get_label* function as follow.

```python
import numpy as np
import torch

# get human label.
# sa_t: the segment; physics_seg: the physics information of the
    segment; env: the environment; video_recorder: the recorder for
    physics rendering.
def get_label(self, sa_t_1, sa_t_2, physics_seg1, physics_seg2, env,
    video_recorder):

    # get human label
    human_labels = np.zeros(sa_t_1.shape[0])
    for seg_index in range(physics_seg1.shape[0]):
        # render the pairs of segments and save the video
        video_recorder[0].init(env, enabled=True)
        for i in range(physics_seg1[seg_index].shape[0]):
            with env.physics.reset_context():
                env.physics.set_state(physics_seg1[seg_index][i])
            video_recorder[0].record(env)
        video_recorder[0].save('seg1.mp4')

        video_recorder[1].init(env, enabled=True)
        for i in range(physics_seg2[seg_index].shape[0]):
            with env.physics.reset_context():
                env.physics.set_state(physics_seg2[seg_index][i])
            video_recorder[1].record(env)
        video_recorder[1].save('seg2.mp4')

        labeling = True
        # provide labeling instruction and query human for preferences
        while(labeling):
            print("\n")
            print("-------------------------------------------------")
            print("Feedback number:", seg_index)

            # preference:
            # 0: segment 0 is better
            # 1: segment 1 is better
            # other number: hard to judge, skip this pair of segment
            while True:
                # check if it is 0/1/number type preference
                try:
                    rational_label = input("Preference: 0 or 1 or other
                        number")
                    rational_label = int(rational_label)
                    break
                except:
                    print("Wrong label type. Please enter 0/1/other number.
                        \n")
```

```
        print("-----------------------------------------------")

        human_labels[seg_index] = rational_label
        labeling = False

# remove the hard-to-judge pairs of segments
cancel = np.where(human_labels!=0 and human_labels!=1)[0]
human_labels = np.delete(human_labels, cancel, axis=0)
sa_t_1 = np.delete(sa_t_1, cancel, axis=0)
sa_t_2 = np.delete(sa_t_2, cancel, axis=0)

print("valid query number:", len(human_labels))
return sa_t_1, sa_t_2, human_labels
```

The user interface is shown in Figure 25. Users have the option to input either 0 or 1 to indicate their preference, or they can input any other number to skip this labeling.

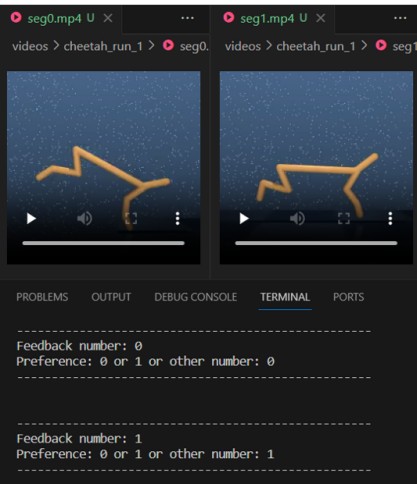

Figure 25: User interface for providing user preferences.

