# OpenReview forum: "Query-Policy Misalignment in Preference-Based Reinforcement Learning"
_ICLR.cc/2024/Conference — ICLR 2024 spotlight_

### Official Review · Reviewer_EPnJ · 2023-11-01

**Soundness:** 3 good
**Presentation:** 3 good
**Contribution:** 3 good
**Rating:** 6
**Confidence:** 3

**Summary:**

Summary: The paper addresses the challenge of Query-Policy Misalignment in Preference-based Reinforcement Learning (PbRL). PbRL is a method where reinforcement learning (RL) agents align their behavior based on human preferences. However, the efficiency of these models is often restricted due to costly human feedback. The paper identifies that most existing PbRL methods aim to improve the reward model's quality by selecting queries but may not necessarily enhance the RL agent's performance. The authors introduce the concept of policy-aligned query and hybrid experience replay as a solution. These methods focus on improving the alignment between the queries chosen and the current interests of the RL agent, thereby enhancing feedback efficiency. This is done by simply sampling from recent trajectories.

**Strengths:**

1. The paper introduces a novel perspective, highlighting the overlooked issue of Query-Policy Misalignment in PbRL.
2. The proposed solution of policy-aligned query selection and hybrid experience replay is simple to implement and requires minimal changes to existing PbRL systems.
3. Comprehensive experiments on well-established benchmarks like DMControl and MetaWorld prove the substantial benefits of the proposed method in terms of feedback and sample efficiency.

**Weaknesses:**

1. The focus is predominantly on off-policy PbRL methods, with limited exploration of on-policy PbRL methods, which naturally select on-policy segments to query preferences.
2. The approach is quite simple and I’m not sure if it’s novel compared to methods that are more similar to on-policy methods.
3. The paper should compare to Liu et al. Meta-Reward-Net: Implicitly Differentiable Reward Learning for Preference-based Reinforcement Learning. NeurIPS ‘22, which is the current SoTA for PbRL.
4. I thought the Figure 1, 2, 3 could be improved and explained better, both in the text, and in the figure, and in the caption.

**Questions:**

Could the proposed methods be adapted or combined with on-policy PbRL techniques to achieve even better results?
Are there specific scenarios or domains where the proposed method may not be as effective?
How does the system handle scenarios where human feedback might be inconsistent or contradictory?

---

> ### Author Response · Authors · 2023-11-15
> **Response to Reviewer EPnJ**
>
> We sincerely thank Reviewer EPnJ for the useful comments. For each weakness and question, we provide the following responses.
>
> > W1. The focus is predominantly on off-policy PbRL methods, with limited exploration of on-policy PbRL methods, which naturally select on-policy segments to query preferences.
>
> - As we discussed in "Conclusion and Discussion" on Page 9, although on-policy PbRL methods naturally select on-policy segments, they suffer from severe sample inefficiency issue as compared to off-policy PbRL methods. Due to the prolonged policy learning process, the query segments generated by slowly learned policies are often not very informative for rapid reward model learning, which in turn causes poor feedback efficiency in on-policy PbRL methods.
> - We add the performance curve of PrePPO (on-policy PbRL method) [1] to Figure 5 and 6 on Page 8. We can see that PrePPO's performance is notably inferior to current SOTA off-policy PbRL methods.
> - All the recent SOTA PbRL methods, including ours, are built upon off-policy framework. We discover that the query-policy misalignment can potentially harm feedback efficiency, and thus enforces query selection to align with the policy induced distribution $d^{\pi}$. Note that this is different from conventional on-policy RL, as our treatment is conducted on the query selection part (i.e., selecting proper query segments that are already in the replay buffer), rather than collecting on-policy samples for policy learning as in typical on-policy RL methods.
>
> > W2. The approach is quite simple and I’m not sure if it’s novel compared to methods that are more similar to on-policy methods.
>
> - To the best of our knowledge, our paper is the first study that reveals the long-neglected query-policy misalignment causes query inefficiency in PbRL. Most existing studies focus on selecting the most "informative" segment for preference labeling. However, as demonstrated in our work, this can be less effective as expected. On the other hand, we provide a remarkably simple method to improve feedback efficiency in PbRL without any fancy query selection scheme.
> - Second, as we have clarified in the previous response, our method is an **off-policy method**, simply using an on-policy PbRL method performs poorly due to sample inefficiency.
>
> > W3. The paper should compare to Liu et al. Meta-Reward-Net: Implicitly Differentiable Reward Learning for Preference-based Reinforcement Learning. NeurIPS '22, which is the current SoTA for PbRL.
>
> - We appreciate the reviewer's helpful reminder! The performance of Meta-Reward-Net (MRN) and its comparison to QPA has been added in Figure 12, Appendix D.2. In most tasks, our method QPA consistently outperforms MRN. Essentially, MRN takes a good idea of adopting the performance of the Q-function as the learning target to formulate a bi-level optimization problem, which is orthogonal to our methods. Our technique *policy-alignment query selection* can be easily incorporated into MRN to further improve feedback efficiency (MRN + policy-alignment query selection).
>
>
> > W4. I thought the Figure 1, 2, 3 could be improved and explained better, both in the text, and in the figure, and in the caption.
>
> - We thank the reviewer for the suggestion. We have added more explanations on Page 2-4. Furthermore, we have provided very detailed experiment settings and descriptions in Appendix C.
>
>
> > Q1. Could the proposed methods be adapted or combined with on-policy PbRL techniques to achieve even better results?
>
> - Our proposed method focuses on the query selection and value learning from the **replay buffer**. In contrast, in on-policy PbRL, these processes occur within **on-policy rollout trajectories**, where the policy-aligned query selection and hybrid experience are no longer applicable. As we have explained in the previous response as well as the additional results of on-policy method PrePPO, the low sample efficiency and prolonged policy learning process in on-policy PbRL can hurt feedback efficiency. By contrast, our proposed method has achieved SOTA performance.
>
> > Are there specific scenarios or domains where the proposed method may not be as effective?
>
> - When the reward function is highly non-smooth in the state-action space and drastically different in various regions, hard exploration might be needed to learn the reward model effectively. In such case, relying solely on policy-aligned query selection might lead to getting trapped in local solutions.
>
> > How does the system handle scenarios where human feedback might be inconsistent or contradictory?
>
> - We have added new experiments with human feedback, which could potentially contain inconsistencies. The results are presented in Appendix E, we find our proposed QPA still achieves reasonable/good performance.
>
> **Reference**
>
> [1] Deep reinforcement learning from human preferences, NeurIPS, 2017.

---

> > ### Comment · Reviewer_EPnJ · 2023-11-21
> > **Reviewer Response**
> >
> > Thanks for the detailed response. I have bumped my score up. Please consider adding the MRN results in the main text.

---

> > > ### Author Response · Authors · 2023-11-23
> > >
> > > Thanks for the good suggestion! We will add the results to the main text in our final version paper.
> > >
> > > We sincerely thank the time and effort you have engaged in the review and discussion phase!

---

### Official Review · Reviewer_EEB5 · 2023-11-01

**Soundness:** 3 good
**Presentation:** 3 good
**Contribution:** 3 good
**Rating:** 6
**Confidence:** 4

**Summary:**

The paper addresses the problem of query-policy misalignment in preference-based reinforcement learning (PbRL) and introduces a novel method, QPA, consisting of two main techniques, policy-aligned query selection, and hybrid experience replay, to improve the efficiency of human feedback in PbRL.

**Strengths:**

* This paper offers a fresh viewpoint on an underexplored issue in PbRL and proposes an interesting solution.

* The conducted experiments are extensive, covering a range of tasks and comparing against multiple benchmark methods.

* The paper is written well, the problem is clearly defined, the proposed approach is thoroughly explained, and the empirical evaluation is meticulously conducted.

* The results show the effectiveness of the proposed method in achieving significant gains in performance.

**Weaknesses:**

* Query-policy misalignment not conclusively proven: While the paper proposes that query-policy misalignment is an interesting hypothesis to cause a problem in PbRL, there is no comprehensive evaluation or solid evidence to confirm this proposal.

* Lack of real human experiments: The testing and validation conducted in the paper relied on a scripted annotator which is not representative of real-world users, which could limit the generalizability and applicability of the approach.

**Questions:**

* The paper could be strengthened by testing the proposed method under real-world conditions, using actual human annotators instead of an oracle, to test the generalizability of their results.

* Providing more implementation details or pseudocode for the proposed method would add value to the paper and make it easier for others to understand, replicate, and build upon the proposed method.

---

> ### Author Response · Authors · 2023-11-15
> **Response to Reviewer EEB5**
>
> We sincerely thank Reviewer EEB5 for the constructive comments. For each weakness and question, we provide the following responses.
>
> > W1. Query-policy misalignment not conclusively proven: While the paper proposes that query-policy misalignment is an interesting hypothesis to cause a problem in PbRL, there is no comprehensive evaluation or solid evidence to confirm this proposal.
>
> - We really appreciate this valuable comment. We have added Figure 11 and Appendix D.1 to confirm this proposal. As all the recent off-policy PbRL methods are built upon PEBBLE, we compare the two methods: PEBBLE and PEBBLE + policy-aligned query selection in locomotion tasks. The policy-aligned query selection is proposed in Section 5.1 to address the query-policy misalignment.
>     - We compute the log-likelihood of current policy $\pi$ using the queried segments at each query time. Figure 11(a) shows that the segments queried by PEBBLE exhibit a low log-likelihood of $\pi$, indicating that these segments fall outside the distribution of the current policy $\pi$. This demonstrates the existence of the query-policy misalignment.
>     - Figure 11(a) shows that, if incorporating policy-aligned query selection into PEBBLE to confine the queries to the local policy-aligned buffer, there will be a substantial increase in the log-likelihood of $\pi$, which means the query-policy misalignment can be addressed.
>     - Figure 11(b) shows that the performance of PEBBLE + policy-aligned query selection significantly surpasses that of PEBBLE.
> - Overall, Figure 11 demonstrates that: (1) the query-policy misalignment issue does exist in typical PbRL methods and does cause feedback inefficiency; (2) using policy-aligend query selection to address the query-policy misalignment can result in a significant improvement in feedback efficiency.
>
>
> > W2. Lack of real human experiments: The testing and validation conducted in the paper relied on a scripted annotator which is not representative of real-world users, which could limit the generalizability and applicability of the approach.
> >
> > Q1. The paper could be strengthened by testing the proposed method under real-world conditions, using actual human annotators instead of an oracle, to test the generalizability of their results.
>
>
> - The use of scripted ground truth reward is to evaluate and compare these PbRL methods quantitatively, unbiasedly and quickly, which is a common practice in existing PbRL literature [1-6].
> - We have included additional real human experiments in Appendix E to showcase the applicability and effectiveness of our method. Using feedback from real humans, our method also notably improves feedback efficiency compared to PEBBLE.
>
>
> > Q2. Providing more implementation details or pseudocode for the proposed method would add value to the paper and make it easier for others to understand, replicate, and build upon the proposed method.
>
> - We thank the reviewer for this kind suggestion. We have provided the pseudocode in Appendix B and implementation details in Appendix C on Page 13-16. These details encompass the implementation framework, query selection scheme, data augmentation explanation, and hyperparameter settings.
>
> **Reference**
>
> [1] Deep reinforcement learning from human preferences, NeurIPS, 2017.
>
> [2] PEBBLE: Feedback-Efficient Interactive Reinforcement Learning via Relabeling Experience and Unsupervised Pre-training, ICML, 2021.
>
> [3] SURF: Semi-supervised Reward Learning with Data Augmentation for Feedback-efficient Preference-based Reinforcement Learning, ICLR, 2022.
>
> [4] Reward Uncertainty for Exploration in Preference-based Reinforcement Learning, ICLR, 2022.
>
> [5] Meta-Reward-Net: Implicitly Differentiable Reward Learning for Preference-based Reinforcement Learning, NeurIPS, 2022.
>
> [6] Inverse Preference Learning: Preference-based RL without a Reward Function, NeurIPS, 2023.

---

> > ### Comment · Reviewer_EEB5 · 2023-11-19
> >
> > Thank you for your detailed answers. I have one more suggestion: it would be nice if the authors could provide more details about the human labeling process (user interface, labeling instruction, when to collect human feedback and so on) for additional human results in Appendix E. Based on responses, I'd like to keep my original rating.

---

> ### Author Response · Authors · 2023-11-19
>
> Thanks a lot for this good suggestion!
>
> As suggested, we have added more experimental details of human labeling process in Appendix E on Page 24-25 in our latest version paper. Specifically, we have provided the main Python code for segment video rendering, labeling instructions, and human preference collection on Page 24-25. The user interface is shown in added Figure 25. The human feedback frequency and total feedback remain consistent with the experimental setup outlined in Table 2. The supplementary material includes the videos of agent training processes in human experiments.
>
> We sincerely thank the time and effort you have engaged in the review and discussion phase!

---

### Official Review · Reviewer_LzS2 · 2023-11-11

**Soundness:** 3 good
**Presentation:** 2 fair
**Contribution:** 3 good
**Rating:** 8
**Confidence:** 4

**Summary:**

In this work, the Authors set to improve feedback efficiency in RLHL (RL from human feedback). To this end, they propose to (1) use only recent behaviors of RL agents for queries and to (2) update RL agents using recent experiences alongside experiences uniformly sampled from the replay buffer. The Authors test their proposed sampling approaches in several existing RLHF models on a number of control benchmarks where they show improved performance in comparison to baseline approaches.

**Strengths:**

The paper addresses an important problem of improving the feedback efficiency in RLHF. As RLHF is widely used today, in particular, with LLMs (large language models), the results offered in this work are important, timely, and definitely relevant to the community.

The simplicity of the approach is also a plus. As the Authors state in the paper, their sampling strategies require minor additions to the existing models’ code, on the order of a couple dozen lines of code, so their approach is easy to implement.

The proposed approach has been tested on a variety of control tasks where it shows improvement in performance compared to the baseline models.

Overall, I think this submission is a solid work with a clear message and thoroughly conducted experiments.

**Weaknesses:**

I wasn’t able to fully follow some of the logic regarding the justification of the sampling scheme design. Throughout Pages 5 and 6, I understood the equations describing the error bounds for the Q-functions and the rewards but I couldn’t see the formal connection between these bounds and the proposed sampling schemes. If there *is* a formal connection, I suggest elaborating on it, e.g. in the statement in 5.1. that goes: “By assigning more observers feedback <…> we aim to enhance the accuracy of the preference (reward) predictor <…> This aligns with the intuition from the condition <…> in Eq. (5).” Similarly, in 5.2, the claim: “The proposed mechanism can provide assurance that the Q-function is updated adequately near $d^\pi$” may need further elaboration formally supporting it. If there *isn’t* a formal connection between these claims and Eq. (5), I suggest removing Eq. (5) and the related references as, in the current standing, they may be confusing to a reader like myself.

At the same time, similar ideas have been explored in literature. The field of decision-aware model learning (or value equivariance model learning) used the idea that, in model-based RL, learning a model that’s accurate everywhere might be unnecessary and, instead, learning a model that’s only accurate in task-critical regions of the state space may offer a way of improving models’ sample complexity. Although these approaches, to my knowledge, have not been applied to RLHF, they pursued similar goals (i.e. improving the sample complexity) by using similar methods (i.e. focusing on the prediction accuracy for task-relevant states and transitions). I suggest discussing these (and similar) lines of work and their relation to the proposed sampling approach in the paper.

**Questions:**

See above

_____________________
Post-rebuttal. The Authors have diligently addressed my concerns and, to my knowledge, did a good job of addressing the other Reviewers' concerns. Also, I see that this paper is of a much higher quality than a typical paper at this venue this year that has been assigned the same score as I initially put here. To reflect on both of these facts, I increase my score.

---

> ### Author Response · Authors · 2023-11-15
> **Response to Reviewer LzS2**
>
> We sincerely thank Reviewer LzS2 for the insightful comments. For each weakness and question, we provide the following responses.
>
> > W1. I wasn’t able to fully follow some of the logic regarding the justification of the sampling scheme design...
>
> - We thank the reviewer for this helpful comment. We explain the connection between the analysis of Eq.(5) and the proposed technique as follows.
>     - The proposed technique, policy-aligned query selection, allocates more queries according to the induced distribution $d^{\pi}$ of current policy $\pi$. This could potentially enhance the accuracy of reward learning within distribution $d^{\pi}$, enforcing the condition of $||\hat{r}\_{\psi} - r||\_{d^{\pi}} \leq \epsilon$.
>     - The proposed technique, hybrid experience replay, samples more fresh transitions within the $d^{\pi}$ to update the Q function. This pays more attention to improving the quality of Q-function within $d^{\pi}$, enforcing the condition of $||Q\_{\hat{r}\_{\psi}}^{\pi} - \hat{Q}\_{\hat{r}\_{\psi}}^{\pi} ||\_{d^{\pi}} \leq \alpha$.
>     - As shown in Eq.(5), these two conditions would lead to a concrete error bound on the approximated Q-value $\hat{Q}\_{\hat{r}\_{\psi}}^{\pi}$.
> - We have taken the reviewer's suggestion and moved the analysis of Eq.(5) to the Appendix A. We thank the reviewer again for this helpful comment.
>
> > W2. At the same time, similar ideas have been explored in literature. The field of decision-aware model learning (or value equivariance model learning) used the idea that, in model-based RL...
>
> - We really appreciate the reviewer for this valuable suggestion! Good point! We have added the discussion about the high-level connections between these model-based decision-making methods and our method in Section 2 on Page 3. This could help the readers to better grasp the core ideas and rationale behind our method.
> - We have added the discussion and refrences of the following model-based decision-making (local decision-aware model learning) papers. Please let us know if we are missing any important relavant works.
>
> **Added relevant model-based decision-making references:**
>
> [1] Learning neural network policies with guided policy search under unknown dynamics, NeurIPS, 2014.
>
> [2] Learning contact-rich manipulation skills with guided policy search, arXiv, 2018.
>
> [3] One-shot learning of manipulation skills with online dynamics adaptation and neural network priors, IROS, 2016.
>
> [4] Sample-based informationl-theoretic stochastic optimal control, ICRA, 2014.
>
> [5] Autonomous helicopter control using reinforcement learning policy search methods, ICRA, 2001.
>
> [6] Locally weighted learning for control, Springer, 1997.
>
> [7] The value equivalence principle for model-based reinforcement learning, NeurIPS 2020.
>
> [8] Value prediction network, NeurIPS, 2017.
>
> [9] The predictron: End-to-end learning and planning, ICML, 2017.
>
> [10] Value iteration networks, NeurIPS, 2016.
>
> [11] Universal planning networks: Learning generalizable representations for visuomotor control, ICML, 2018.
>
> [12] Live in the moment: Learning dynamics model adapted to evolving policy, ICML, 2023.
>
> [13] Objective mismatch in model-based reinforcement learning, L4DC, 2020.
>
> [14] Learning Policy-Aware Models for Model-Based Reinforcement Learning via Transition, L4DC, 2023.

---

> > ### Comment · Reviewer_LzS2 · 2023-11-20
> >
> > Thank you for your response. This addresses my questions. I also appreciate highlighting the text changes in blue which made them easy to follow.
> > -Re: prior literature: nice reference list! This definitely includes the papers I had in mind;
> > -Re: motivation of the method: thanks for adding clarity on the motivation being an *intuitive* explanation. As long as it's stated as such, I believe it's completely correct.

---

> > > ### Author Response · Authors · 2023-11-20
> > >
> > > We really appreciate the time and effort you have engaged in the review and discussion phase! These valuable comments help us improve our work significantly!

---

> > > ### Author Response · Authors · 2023-11-21
> > > **Post-rebuttal response**
> > >
> > > We are glad that our work has received your recognition! Thanks again for your valuable comments and suggestions!

---

### Author Response · Authors · 2023-11-15
**General Response**

We thank all the reviewers for the effort engaged in the review phase. Regarding the concerns of the reviewers, we have revised our paper (highlighted in blue text color) and summarized the modifications in the following.
- (For Reviewer LzS2) We have removed Eq.(5) in Section 4 and moved the analysis of Eq.(5) to the Appendix A on Page 13.
- (For Reviewer LzS2) We have added the related work on local decision-aware model learning in Section 2 on Page 3.
- (For Reviewer EEB5) We have added more evidence to confirm the feedback inefficiency caused by query-policy misalignment in Figure 11 and Appendix D.1 on Page 17-18.
- (For Reviewer EEB5) We have added additional real human experiments in Appendix E on Page 22-23.
- (For Reviewer EEB5) We have provided the pseudocode in Appendix B and implementation details in Appendix C on Page 13-16.
- (For Reviewer EPnJ) We have added the performance of PrePPO (on-policy PbRL method) in Figure 5 and 6 on Page 8.
- (For Reviewer EPnJ) We have added the performance of Meta-Reward-Net (MRN) and its comparison to QPA in Figure 12, Appendix D.2 on Page 17-18.

---

### Meta-Review · Area_Chair_eCih · 2023-12-07

**Metareview:**

The paper studies RLHF for aligning agents' behaviors to human preferences. The authors propose novel query selection and experience replay schemes to improve the data efficiency of RLHF. All the reviewers appreciated that the paper contributes novel, effective solutions for a timely problem, and that the authors conducted a thorough evaluation (including a human user study during the author feedback phase). Hence, all reviewers agree that the paper is above the bar for publication.

**Justification For Why Not Higher Score:**

Three factors can make the paper's contributions even stronger: (1) discussion, comparison and extension of the proposed ideas to on-policy preference-based learning methods also; (2) Some reviewers pointed out how the introduction and initial figures can be better explained for a better exposition; and (3) providing code implementations and detailing the human annotation process to ensure replicability of the findings and adoption by the community.

**Justification For Why Not Lower Score:**

The major concerns raised by the reviewers were effectively addressed by the authors during the feedback phase, including additional experiments, appendices with additional details, etc.

---

### Decision · Program_Chairs · 2024-01-16

Accept (spotlight)